# Rational engineering of allosteric protein switches by in silico prediction of domain insertion sites

Benedict Wolf [1], Pegi Shehu[1,2], Luca Brenker[1], Anna-Lisa von Bachmann[1,3], Ann-Sophie Kroell[1], Nicholas Southern[1], Stefan Holderbach [1], Joshua Eigenmann [1], Sabine Aschenbrenner[1], Jan Mathony [1,4] ✉ & Dominik Niopek [1,4] ✉

Domain insertion engineering is a powerful approach to juxtapose otherwise separate biological functions, resulting in proteins with new-to-nature activities. A prominent example are switchable protein variants, created by receptor domain insertion into effector proteins. Identifying suitable, allosteric sites for domain insertion, however, typically requires extensive screening and optimization. We present ProDomino, a machine learning pipeline to rationalize domain recombination, trained on a semisynthetic protein sequence dataset derived from naturally occurring intradomain insertion events. ProDomino robustly identifies domain insertion sites in proteins of biotechnological relevance, which we experimentally validated in *Escherichia coli* and human cells. Finally, we used light- and chemically regulated receptor domains as inserts and demonstrate the rapid, model-guided creation of potent, single-component opto- and chemogenetic protein switches. These include novel CRISPR–Cas9 and –Cas12a variants for inducible genome engineering in human cells. Our work enables one-shot domain insertion engineering and substantially accelerates the design of customized allosteric proteins.

Domains are structural and functional subunits of proteins[1]. The rearrangement of existing domains into new combinations is a major source of evolutionary innovation[2]. While individual domains are by definition structurally and functionally conserved, they naturally occur in various protein contexts[1–3]. Artificial recombination of protein domains has proved to be a powerful method for engineering novel proteins with properties not found in nature (Fig. 1a)[4,5]. To achieve not only a physical connection but also an effective functional coupling of two naturally separated protein domains, one particularly powerful strategy is to insert a domain into a selected site of another. If successful, this results in a tight structural and functional codependence of the fused domains.

This engineering strategy has great potential for the development of sensitive biosensors and novel molecular machines, as best exemplified by engineered, allosteric protein switches[6,7]. More specifically, the insertion of a photoreceptor or ligand-binding domain into an acceptor protein of choice can yield switchable proteins whose activity is regulated by light or small molecules. In these cases, a trigger-mediated conformational change of the receptor domain is propagated to the acceptor protein, thereby activating or deactivating it, as previously shown by us and others[6,8,9].

Progress in the field of engineering allosteric protein effectors is, however, largely hampered by the difficulty of identifying suitable

[1]Institute of Pharmacy and Molecular Biotechnology, Faculty of Engineering Sciences, Heidelberg University, Heidelberg, Germany. [2]Department of Biology, Technical University of Darmstadt, Darmstadt, Germany. [3]Zuse School ELIZA, Darmstadt, Germany. [4]These authors jointly supervised this work. ✉e-mail: jan.mathony@uni-heidelberg.de; dominik.niopek@uni-heidelberg.de

**Fig. 1 | Intradomain insertions are common in natural proteins. a**, Novel proteins can arise from the insertion of one domain into another. **b**, Strategy for generating a large domain insertion dataset of natural proteins. **c**, The number of unique domain superfamily combinations is shown for a given parent (left panel) or insert domain (right panel). Only the top five most promiscuous domains are shown. **d**, Alphafold2-generated structures of example proteins from the dataset with PDZ domain insertions. Insert domains are in green and parent proteins in blue. **e**, Length distribution of insert domains, parent domains (without the respective insert) and parent proteins in the dataset. Parent domains are defined as the annotated Interpro domains originally carrying the insert, while parent protein refers to the full-length protein without the insert. aa, amino acids. **f**, Distribution of relative insertion site positions within parent domains and parent proteins. **g–j**, The frequency of different CATH-GENE3D domain types at the 'class' (**g**) and 'architecture' (**h**, mainly alpha; **i**, mainly beta; **j**, alpha beta) levels (according to the CATH hierarchy) within the whole CATH database are compared with the corresponding distribution of the insert and parent domains in our dataset.

domain insertion sites in effector proteins, as a naive domain fusion usually results in permanent impairment of the structural and functional integrity of one or both domains[10–12]. In addition, while the few sites in a protein that tolerate domain insertion are often promiscuous with respect to the type of insert domain, several studies have shown that their rational inference or computational prediction is challenging[10,13]. Although surface-exposed flexible loops are often considered promising insertion sites, experimental screens have shown that only a fraction of these loops actually tolerate domain integration

and that secondary structure elements in turn can sometimes accept insertions very well[6,10,11]. Finally, even the application of simple, basic constraints such as surface exposure are prone to misguided selection of insertion sites, likely due to our limited understanding of conformational ensembles and structural features that confer protein activity in the native, cellular environment.

The main limitation in identifying sites suitable for domain fusion is the scarcity of informative data, as experimental domain insertion datasets are cumbersome to generate and exist only for few

proteins. While we and others have trained random forests and similar machine learning algorithms on insertion mutagenesis datasets, these relatively simple models are unable to generalize to unrelated protein families and are therefore not suitable for actually guiding domain insertion engineering[10,13,14]. Protein language models on the other hand, have revolutionized protein engineering by enabling accurate prediction of protein structure[15,16] or functional properties[17–19] based on feature-rich representations and have been successfully applied to various protein engineering and design tasks[20,21]. However, until now, these data-intensive algorithms have not been applicable to the problem of domain insertion engineering due to the lack of suitable training data.

In this study, we developed a robust machine learning-based approach to guide the engineering of protein domain recombination. Instead of relying on sparse experimental data, we constructed a large protein dataset consisting of semisynthetic sequences inspired by naturally occurring intradomain insertion events. This diverse dataset enabled us to train ProDomino (protein domain insertion optimizer), a powerful model for predicting domain insertion sites in effector proteins. Using ESM-2-derived protein sequence representations as model inputs in combination with a masking strategy, we were able to fine-tune the sensitivity and selectivity of the pipeline to achieve accurate insertion site prediction in proteins evolutionarily unrelated to the training data. Applying ProDomino, we inferred insertion sites for several proteins with success rates of around 80% as validated by experiments in *E. coli* and human cells. Finally, by inserting receptor domains responsive to blue light or chemical inducers, we demonstrated the direct, model-based engineering of several switchable effector proteins of biotechnological and biomedical relevance. These include two common antibiotic resistance-mediating enzymes as well as the widely employed genome editors CRISPR–Cas9 and –Cas12a. Our pipeline enables the creation of switchable, allosteric effector proteins and paves the way for domain recombination engineering at scale.

## Results

### Curating an artificial domain insertion dataset
The use of in silico approaches to identify domain insertion sites in proteins is currently limited by the lack of sufficiently large and diverse datasets. We sought to create a new dataset informative toward insertion tolerance and domain–domain communication by focusing on proteins in which domains are tightly coupled at the structural and functional level.

We speculated that 'intradomain insertions' that occur in natural proteins at low frequency, that is cases in which one domain has been evolutionarily acquired by another domain, resulting in hybrid domain architectures, may provide the information we seek for domain recombination engineering. In contrast, conventional head-to-tail domain concatenations may not carry the same information, as they are expected to be much less structurally constrained and these domains may rather act as functionally independent entities.

Toward curating a dataset of intradomain insertions, we used CATH-Gene3D domain annotations, which are defined as structural superfamilies[22,23], from Interpro[24,25]. These were filtered for proteins, in which one domain annotation is interrupted by that of a second domain, thus representing cases in which the N- and C-terminal part of a domain A are separated by a second domain B (Fig. 1b). The remaining sequences were intersected with UniRef50 (ref. 26) to limit pairwise sequence identities to a maximum of 50%, resulting in a final dataset of 174,872 sequences across the tree of life.

The curated set includes 202 distinct insert domain superfamilies and 168 unique parent domain types, most of which occur only as either insert or parent. The promiscuity of both insert and parent domains differs substantially between superfamilies (Fig. 1c and Supplementary Fig. 1). For example, the P-loop NTPase domain serves as a parent domain in combination with 13 different insert superfamilies.

The most promiscuous insert domain, in turn, is the PDZ superfamily (Fig. 1d), which occurs in combination with 11 different parents. Most domains, however, are represented only in combination with one or two other domains (Supplementary Fig. 1). The size distributions of the parent and insert domains differ only slightly, with median sequence lengths of ~160 and ~120 amino acids, respectively, whereas the length of the entire parent proteins is more diverse (Fig. 1e). The relative insertion site within the parent domains is bimodally distributed, presumably because insertions in the middle of a domain are more likely to severely disrupt its function (Fig. 1f). Relative to the full-length parent proteins, the insertion site distribution shows a C-terminal bias, a trend previously observed in the context of a much smaller dataset (Fig. 1f)[27].

Of note, no big differences were observed between the overall domain frequencies in the CATH-Gene3D database and their representation as inserts or parents in our dataset (Fig. 1g–j), suggesting that intradomain insertions are a universal phenomenon across the protein landscape and not restricted to specific domain superfamilies. Similarly, the analysis of residues surrounding domain insertion sites, split by CATH classes, did not reveal any notable enrichment of insertion site-specific sequence motifs (Supplementary Fig. 2). In summary, intradomain insertions are a widespread natural phenomenon, resulting in a diverse dataset with no apparent bias toward specific domain classes.

### ProDomino accurately predicts domain insertion tolerance
Using this dataset as a basis, we trained machine learning models for the prediction of domain insertion sites in proteins. Details on training data and model training procedures are provided in Fig. 2a, Supplementary Note 1 and Supplementary Figs. 3 and 4. In brief, we artificially removed the insert domains from the sequences in our dataset. We then assigned the corresponding positions as positive labels (insertion tolerant), whereas all other positions were assigned negative labels (unknown). We compared several sequence encoding strategies, including simple one-hot encodings and ESM-2-derived embeddings (Fig. 2b), as well as model architectures ranging from smaller multilayer perceptrons to large transformer models (Supplementary Fig. 4)[16]. Moreover, we used different dataset splits: a random split ('random'), a split based on domain classes ('interpro') or a highly strict split using only single representatives per class ('single') (Fig. 2c). Finally, during loss computation, we opted to use a positional masking strategy to account for imbalances within the dataset (Fig. 2d). Using ESM-2 sequence embeddings in combination with the very strict, 'single' split and the masking strategy enabled us to train a performant predictor named ProDomino, well-capable of generalizing beyond the domain insertions observed in the training dataset (Supplementary Note 1).

Analysis of ProDomino predictions revealed the absence of any preference for specific secondary structure motifs (Fig. 2e). When examining selected proteins with available structure information, no strong bias toward surface exposure or sequence conservation was found (Supplementary Fig. 5). Most importantly, the true insertion sites were correctly identified and high scoring positions occurred at frequencies consistent with previous experimental findings (Fig. 2f,g)[10,11]. Notably, the predictive stringency of the model can be fine-tuned over a wide range by varying the number of training steps (Extended Data Fig. 1). Longer training regimes led to stricter classification, that is, fewer peaks with high scores, but also lowered the overall sensitivity.

To benchmark ProDomino on experimental data, we focused on the bacterial transcription factor AraC, which we had comprehensively screened for domain insertion tolerance in a previous study[10]. ProDomino predicted high insertion scores predominantly for experimentally validated insertion sites and correctly identified sequence stretches that did not tolerate insertions (Fig. 2h, Extended Data Fig. 2 and Supplementary Fig. 6), as indicated by a large area under the receiver operator characteristic (AUROC) of 0.84 (Fig. 2i). As expected,

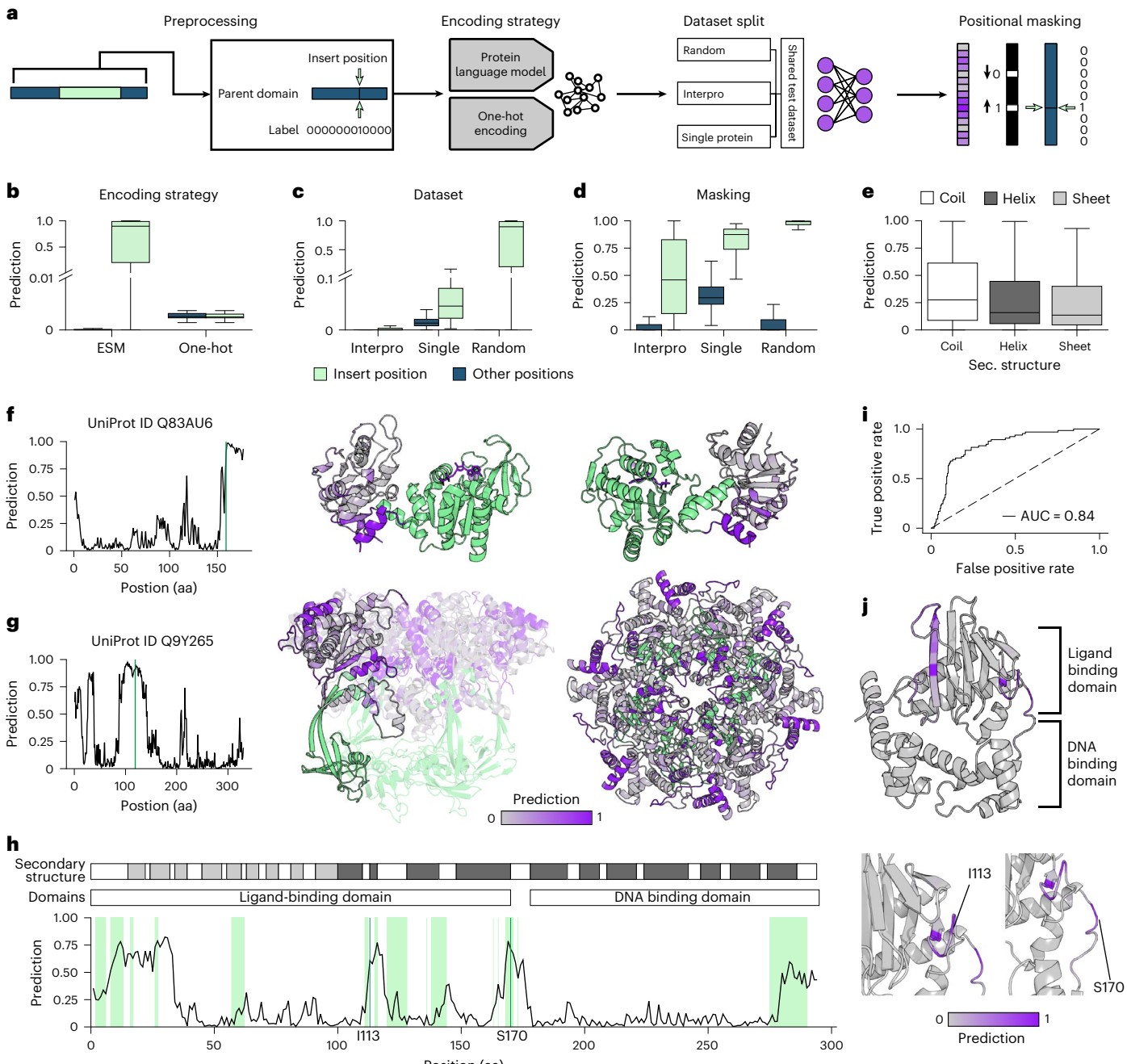

**Fig. 2 | A machine learning model to infer domain insertion sites in proteins.**
**a**, Schematic of the machine learning pipeline for protein insertion site prediction.
**b–d**, Boxplots showing prediction scores for true positive (green) and negative
labels (other positions, unknown; blue) on a test set. The performance of different
models trained with different encoding strategies (**b**), on different dataset splits
(random; Interpro, based on domain classes; single, one representative example
per class) (**c**) or using positional masking (**d**) is shown (see Supplementary
Note 1 for details). **e**, Boxplot of insertion scores predicted by the model variant
trained on the 'single' representative protein dataset split, grouped by secondary
(Sec.) structures. The calculation is based on secondary structure predictions
for the entire test set. **b–e**, Boxes represent the interquartile range (IQR) and
the median is represented by a horizontal line. Whiskers extend to the 1.5-fold

IQR or to the value of the smallest or largest predicted value. *n* = 1,382 protein
sequences with 1,382 known positive insertion sites and 325,510 unknown sites.
**f,g**, Exemplary predictions from the test set. The natural insertion sites are marked
in green and the insert domain is colored accordingly in the protein structures.
**f**, Phosphoglycerate kinase (PDB ID 4NG4); **g**, Rvb1/Rvb2 heterohexamer (RuvB-like 1,
PDB ID 5OAF). **h**, The insertion score for the bacterial transcription factor AraC is
indicated for each amino acid position by a black line. Green positions indicate
experimentally validated insertion-tolerant sites[10]. The domains and secondary
structure elements of AraC are annotated. **i**, AUROC plot of the insertion site
prediction for AraC. **j**, Insertion scores mapped onto the Alphafold2-predicted
AraC protein structure. In **h,j**, allosteric insertion sites, previously validated in
experiments (I113 and S170), are indicated. AUC, area under the curve.

model performance varied depending on how long the model had
been trained, as indicated by a strong drop after 3,000 training steps
(Extended Data Fig. 2 and Supplementary Fig. 6). We therefore decided
to use ProDomino in its 1,500 training step version.

Finally, we had previously identified two allosteric sites in
AraC, that is at position I113 and S170, that resulted in potent light-
dependent protein switches on insertion of the *Avena sativa* LOV2
photoreceptor domain (*As*LOV2) (Fig. 2h–j, Extended Data Fig. 2 and

Supplementary Fig. 6)[10]. ProDomino successfully predicted both regions as the two main peaks with high insertion scores, indicating its potential to identify not only insert-tolerating, but actually allosteric surface sites in proteins (Supplementary Fig. 6). We also compared ProDomino to a previous gradient boosting classifier model inferring domain insertion sites from experimental data and found it to be superior (Supplementary Fig. 7; see Supplementary Note 2 for details). Finally, as additional independent benchmarking, we used previously reported domain insertion datasets for the Kir2.1K$^+$ ion channel[11] and engineered interlaid deaminase domain CRISPR base editors[28–31] We observed that the ProDomino prediction matched the experimental data well overall (Supplementary Fig. 8).

### ProDomino identifies allosteric insertion sites in enzymes

Having validated ProDomino on available experimental domain insertion data, we intended to apply it for engineering switchable single-chain protein effectors. As a starting point, we applied ProDomino to two commonly used enzymes mediating antibiotic resistance in mammalian cells and bacteria, namely puromycin acetyltransferase (PAC) and chloramphenicol acetyltransferase (CAT) (Fig. 3a,b). The highest insertion scores were predicted around E83 (PAC) and K136 (CAT), both located at surface sites in or adjacent to α-β connecting loops (Fig. 3c,d and Supplementary Fig. 9). We introduced the *As*LOV2 photosensory domain behind the corresponding residues and experimentally evaluated the activity of the resulting hybrid PAC-LOV2 and CAT-LOV2 proteins (Fig. 3e).

PAC activity was assessed by expression in human embryonic kidney 293T (HEK293T) cells, followed by treatment with puromycin and illumination of samples with blue light or incubation in the dark for 48 h. Microscopy revealed that cells expressing the PAC-LOV2 hybrid showed wild-type-like puromycin tolerance when incubated in the dark, but died under illumination within 48 h, indicating potent, light-mediated inhibition of PAC (Fig. 3f). In contrast, control samples expressing no PAC or wild-type PAC died or survived the puromycin treatment, respectively, regardless of illumination (Fig. 3f). Complementarily, we evaluated the behavior of two CAT-LOV2 hybrids only differing in their inter-domain linker sequences, by measuring cell growth of *E. coli* cultures supplemented with chloramphenicol and incubated under light and dark conditions. Again, both hybrid enzymes acted as highly potent, light-dependent kill switches, with a 20-fold difference in optical density of the respective cultures between dark and light growth conditions (Fig. 3g and Supplementary Fig. 10). Control cultures expressing wild-type CAT or no CAT at all grew normally or died in the presence of chloramphenicol, regardless of illumination. Finally, illuminating red fluorescent cells expressing the CAT-K136-GS-LOV2 variant on chloramphenicol agar plates with a blue light pattern restricted cell growth to nonilluminated plate areas, indicating precise spatial control of cell killing (Fig. 3h and Supplementary Fig. 11).

Encouraged by the highly successful prediction of allosteric insertion sites in both enzymes, we selected several additional sequence positions with either high or low insertion scores to more globally validate ProDomino's accuracy. To avoid overestimating model performance, we did not include the N- and C-terminal sites, as protein termini are expected to tolerate domain fusions rather well and it is not surprising that a model would pick up this trend. After insertion of the *As*LOV2 receptor or an inert PDZ domain at the selected positions within the PAC and CAT (Extended Data Fig. 3a,c), we screened the resulting hybrid proteins using an MTT (a yellow tetrazole) cell viability assay in mammalian cells or *E. coli* culture growth as readouts. Three out of four low scoring sites in PAC and five out of six low scoring sites in CAT did not tolerate domain insertions (Extended Data Fig. 3b,d). Similarly, most of the positively predicted sites resulted in active fusion proteins (Extended Data Fig. 3b,d). Domain insertion into PAC between S81 and A86, which represents the global prediction maximum, resulted in functional protein hybrids at all positions. Regions with moderate scores (S101-P121), in turn, resulted in lower antibiotic resistance, indicating reduced PAC activity (Extended Data Fig. 3b). In the same line, two out of three CAT positions with high insertion scores tolerated PDZ and/or LOV2 fusion (S27/T36, K136/G137). The only false positive position (A161/N162) is located at the interface of CAT monomers in the homo-trimeric CAT complex; however, no information about the trimeric state of the CAT enzyme was provided to ProDomino. Rerunning the prediction by entering a trimeric CAT sequence, that is an artificial concatemer of three CAT sequences, resulted in an insertion prediction trace without the false positive peak at A161/N162 (Supplementary Fig. 12).

Overall, experiments confirmed model prediction in 78% of cases, even when consecutive, insertion-tolerant sites were counted as a single hit, with a particularly high success rate for positive predictions. Together, these findings indicate that ProDomino is well suited to guide receptor insertion engineering, enabling straightforward creation of optogenetic effector proteins.

### Creating opto- and chemogenetic CRISPR genome editors

To assess the performance of ProDomino on complex, multidomain proteins uniting various functionalities, we finally applied it to infer allosteric sites in CRISPR–Cas nucleases, the workhorses for genome engineering. We started with the most widely used CRISPR effector, Cas9 from *Streptococcus pyogenes*, and first compared our insertion site predictions with a previously reported, experimental transposon insertion dataset. Despite considerable overlap between the literature data and our model prediction, as indicated by an AUROC of 0.71 (Fig. 4a,b and Supplementary Figs. 13 and 14)[32], ProDomino predicted several regions in Cas9 with high insertion scores that, were not among the top hits of the previously published dataset (Extended Data Fig. 4). To dissect this apparent discrepancy, we inserted the *As*LOV2 at four of these 'conflicting' sites into a catalytically impaired d(ead)Cas9. We then expressed the resulting dCas9-LOV2 hybrids, fused to a VPR transactivator, with a luciferase reporter and a promoter-targeting single-guide RNA (sgRNA) in HEK293T cells. Samples were then incubated under blue light or in the dark for 48 h, followed by luciferase assay (Fig. 4c). In line with our predictions, all four dCas9-LOV2 hybrids strongly induced luciferase expression in the dark at levels comparable to the wild-type dCas9-VPR control, indicating high insertion tolerance. Three of the four variants were highly photosensitive, resulting in high or low luciferase activity in the dark or light, respectively. A dCas9/VPR control without the LOV2 insert showed potent reporter activation, independent of illumination (Fig. 4c).

Finally, we sought to apply ProDomino to engineer a switchable, single-chain CRISPR–Cas12a, starting from the *Moraxella bovoculi* (*Mb*) ortholog[33,34], with the goal of rendering it responsive to light or a clinically relevant drug. Ligand-activated, single-chain Cas12a switches would have various potential applications in biotechnology and biomedicine but were thus-far difficult to create.

We passed the *Mb*Cas12a sequence through our ProDomino pipeline. Insertion probability scores across the sequence showed a much more rugged shape with multiple maxima compared to previous predictions (Fig. 4d–f). This suggests Cas12a to be rather promiscuous in its tolerance to domain fusion. We decided to first test several high scoring regions for PDZ insertion tolerance and additionally included eight randomly selected low scoring sites as controls. To assess insertion tolerance, we used a reporter assay in HEK293T cells in which catalytically active Cas12a interferes with luciferase expression via targeted reporter plasmid cleavage. All tested Cas12a hybrids that were predicted to be functional by ProDomino mediated strong reporter knockdown that was in most cases comparable in efficiency to the wild-type nuclease (Extended Data Fig. 5a,b). In contrast, 50% of the Cas12a-PDZ hybrids based on low scoring sites turned out to be dysfunctional, with the remainder still

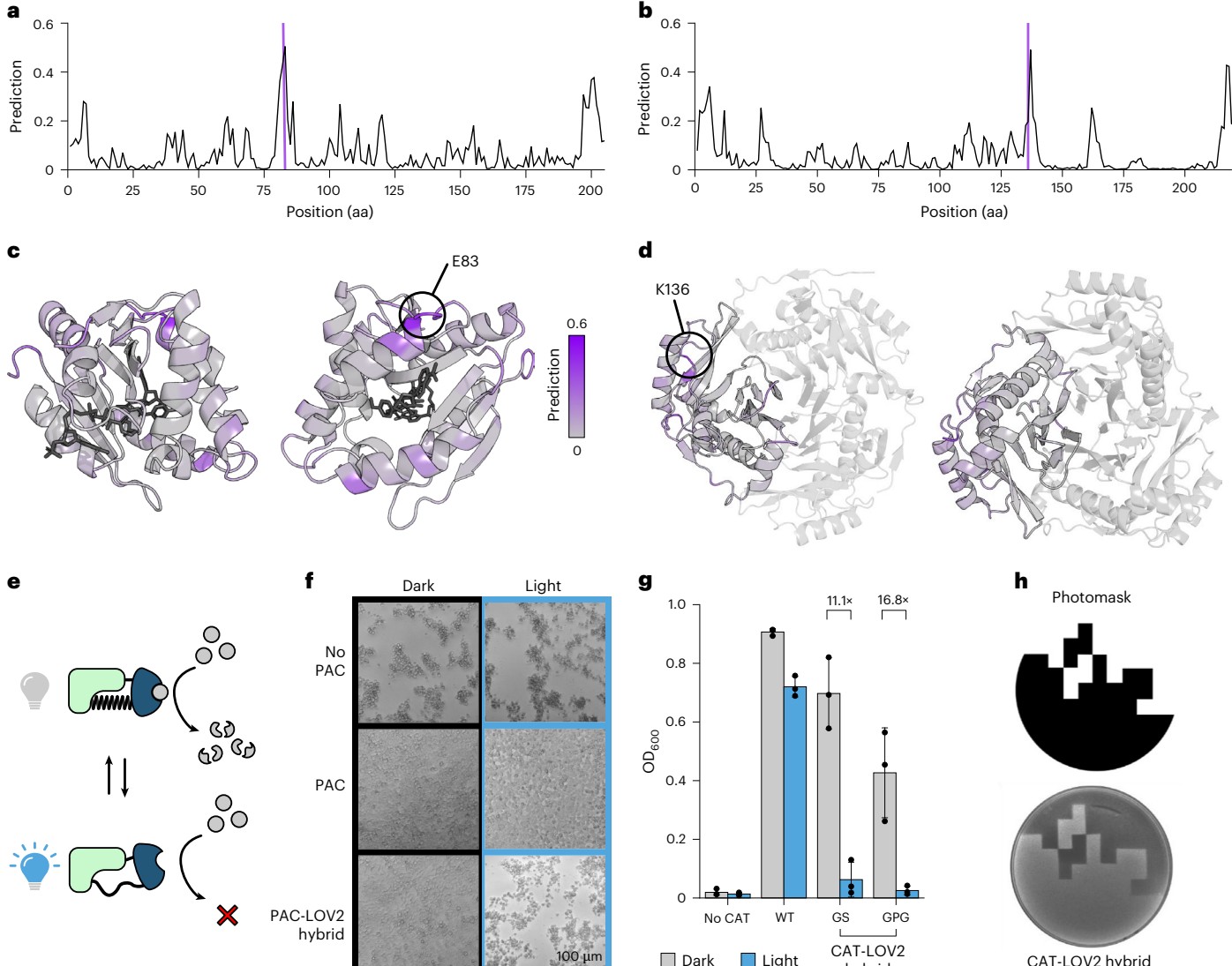

**Fig. 3 | ProDomino informs the engineering of light-controlled antibiotic resistances. a**,**b**, The insertion scores predicted by ProDomino are mapped onto the primary sequences of PAC (**a**) and CAT (**b**). Sites selected for experimental testing by domain insertion are marked with a purple line. **c**,**d** Insertion scores are mapped onto the crystal structures of PAC (**c**) and CAT (**d**). The selected insertion sites are indicated (PDB ID 7K0A and 1PD5). **e**, Scheme of light-regulated PAC/CAT function. **f**, Light control of puromycin resistance. HEK293T cells transfected with vectors encoding the respective PAC variants or a negative control expressing enhanced green fluorescent protein (eGFP, control) were treated with 10 µg ml⁻¹ puromycin, starting 24 h posttransfection. Illumination (or incubation in the dark) began concurrently with the start of puromycin treatment and continued for 48 h, followed by microscopy. The experiment was independently replicated three times under similar conditions and a representative image is

shown. **g**, Light-controlled *E. coli* culture growth. Bacteria were transformed with plasmids expressing the indicated CAT variant or an empty control plasmid. Liquid cultures were grown in the presence of 25 µg ml⁻¹ chloramphenicol and exposed to blue light for 7 h or kept in the dark, followed by assessment of cell density at 600 nm. Bars indicate means, error bars the standard deviation and black dots individual data points from *n* = 3 independent experiments. Amino acid sequences of the symmetric linkers at the receptor domain boundaries are indicated. GS, glycine–serine linker; GPG, glycine–proline–serine linker; WT, wild type. **h**, Spatial regulation of bacterial growth. *E. coli* expressing CAT-K136-LOV and monomeric red fluorescent protein (mRFP) were plated in top agar supplemented with 25 µg ml⁻¹ chloramphenicol. During incubation at 37 °C, the plates were illuminated through a photomask (top) and fluorescent cells were imaged under UV light (bottom).

tolerating the insertion (Extended Data Fig. 5c). These observations underscore that ProDomino is highly reliable in predicting sites for domain fusion, while low scores are not always accurate, at least regarding general domain tolerance.

Next, we selected the two most active hybrid variants carrying insertions at K487 and N1153 for downstream engineering of switchable Cas12a. K487 is located at the interface between the NUC and REC lobes, whereas N1153 is located in the catalytic RuvC domain (Supplementary Fig. 15)[33]. We replaced the PDZ domain at these sites with the *As*LOV2 photosensor and tested the resulting hybrids by targeting the endogenous *RUNX* and *VEGFA* loci under light and dark conditions.

The Cas12a-LOV2 hybrid based on the N1153 insertion site showed potent, light-dependent genome editing, with a threefold reduction in Cas12a activity in the light condition, as validated by T7 Endonuclease I (T7E1) assay and deep amplicon sequencing (Supplementary Fig. 16).

Finally, we sought to enable the control of the allosteric Cas12a variant with a clinically relevant drug instead of light, while also inverting the mode of action from trigger-impaired to trigger-activated. To this end, we employed a circularly permuted variant of the human glucocorticoid receptor 2 (GR2) ligand-binding domain[35,36] that adopts a compact fold in the presence of its ligand cortisol. This is in contrast to the light-dependent unfolding characteristic of *As*LOV2. Following

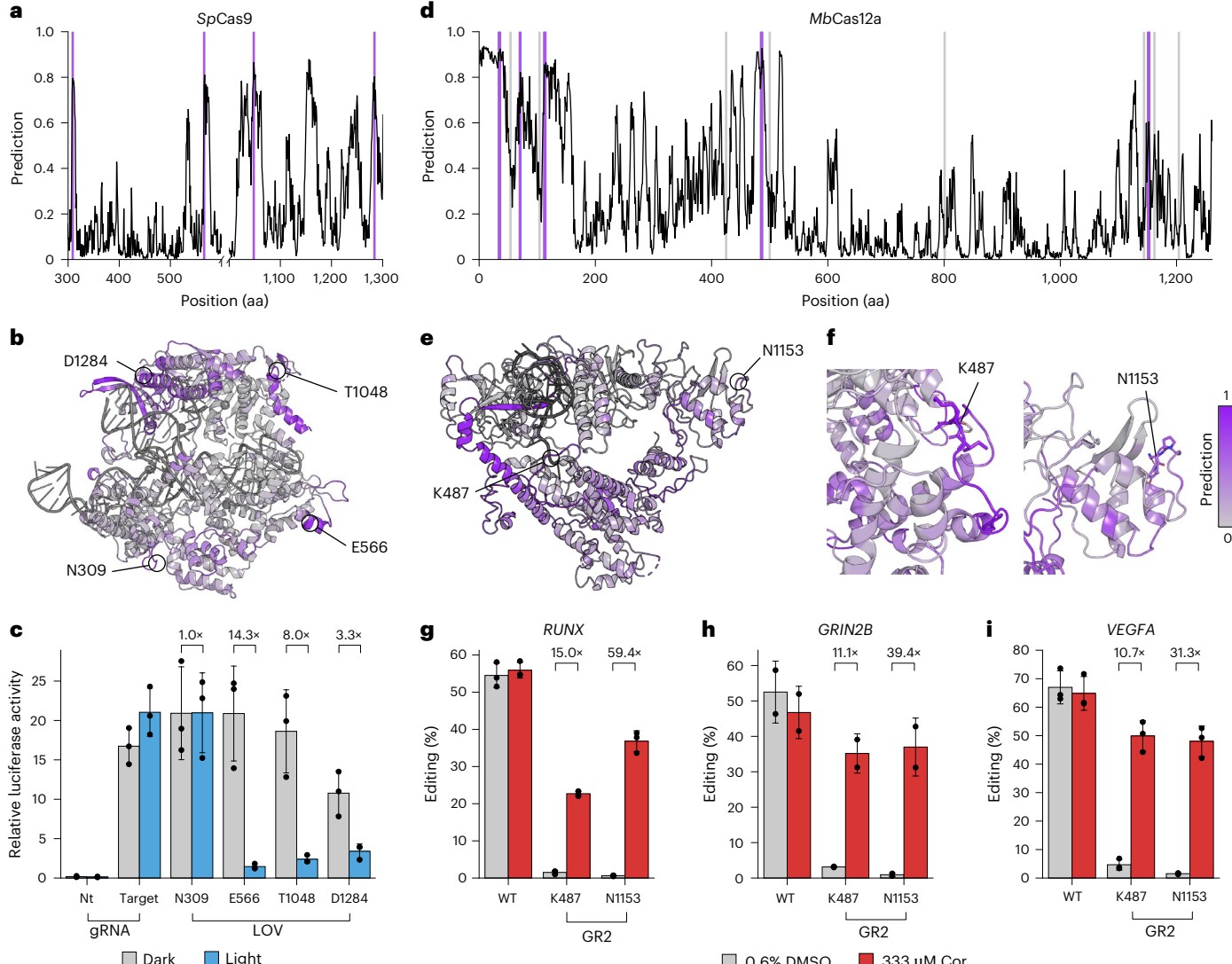

**Fig. 4 | ProDomino confidently predicts potent opto- and chemogenetic Cas9 and Cas12a variants. a,d,** Insertion scores predicted by ProDomino are mapped onto the primary sequences of Cas9 (**a**) or Cas12a (**d**). Selected high- and low scoring sites are marked in purple and gray, respectively. **b,e,** The insertion scores predicted by ProDomino are mapped onto experimentally resolved structures of Cas9 (**b**) and Cas12a (**e**). Insertion sites selected for experimental validation are indicated (PDB ID 4UN3 and 6IV6). **f,** Zoomed-in views of the insertion sites of the two Cas12a lead candidates. **c,** HEK293T cells were transfected with vectors encoding (1) the indicated Cas9/VPR-LOV hybrid variant (or Cas9/VPR as control), (2) a TetO targeting sgRNA together and a *Renilla* luciferase and (3) a firefly luciferase preceded by multiple TetO repeats. Samples were incubated under blue light or in the dark for 48 h, followed by luciferase assay. **g–i,** HEK293T cells were transfected with vectors encoding (1) the indicated Cas12a–GR2 hybrid variant (or wild-type Cas12a as control), and (2) a gRNA targeting the endogenous *RUNX* (**g**), *GRIN2B* (**h**) or *VEGFA* (**i**) locus. Samples were treated with cortisol or DMSO as indicated. 72 h posttransfection, InDel frequencies were assessed by next-generation sequencing. **c,g–i,** Bars indicate means, error bars the standard deviation and black dots individual data points from *n* = 2 (**h**) or *n* = 3 (**c,g,i**) independent experiments. Cor, cortisol.

GR2 insertion at K487 or N1153, we expressed the Cas12a–GR2 hybrids in HEK293T cells and targeted multiple genomic loci, or alternatively a luciferase cleavage reporter. We observed potent, cortisol-induced gene editing at all target sites for both insertion variants (Fig. 4g–i and Supplementary Fig. 17). The N1153 insertion variant showed editing frequencies around 70% of the wild-type enzyme in presence of the ligand, while editing in the absence of cortisol dropped to levels approaching the detection limit (Fig. 4g–i and Supplementary Fig. 17). Control samples expressing wild-type Cas12a showed potent gene editing (InDel frequencies around 50–70%), irrespective of the chemical trigger.

Collectively, these data demonstrate the power of ProDomino for engineering allosteric variants of complex proteins and showcase Cas12a–GR2 hybrids as potent switches for controlling genome editing in human cells.

## Discussion

We introduced ProDomino, a powerful computational model for the prediction of domain recombination and low-N engineering of allosteric protein switches. Key to our approach was the creation of a semi-synthetic domain insertion dataset, which enabled us to train models capable of generalizing beyond the training data[10,11]. The training objective differed from machine learning approaches common to protein engineering problems, due to the lack of true negative samples in our case. As a result, for the problem to be successfully addressed, we could not fully rely on conventional model performance metrics but observed that it was critical to benchmark our model on real-world experimental data early on to refine the training regime (Fig. 2c,d,h and Extended Data Figs. 1 and 2). We found that positional masking in combination with relatively short training regimes could effectively compensate for

imbalances in the dataset, resulting in successful training of a predictive model: ProDomino. Our thorough experimental validation on four proteins from three structurally and evolutionarily unrelated families highlighted the predictive capabilities of ProDomino.

A particularly notable success of ProDomino has been its ability to guide the engineering of switchable effector proteins, which we believe will enable future research and hold great potential for applications in research and biomedicine. The resulting effector variants exhibited robust response to the chemical or light input, without any further optimization. Our engineered genome editors enabled tight control of gene targeting with dynamic regulation ranges unmet thus-far for single-chain, allosteric CRISPR–Cas switches. The Cas12a receptor hybrids in particular greatly enhance the versatility of on-demand inducible CRISPR genome editing.

While ProDomino focuses on predicting domain insertion sites in effector proteins, the choice of the insert domain (receptor) is equally crucial for engineering success. Our previous observations suggest that using larger insert domains tends to negatively affect insertion tolerance[10]. Beyond size, the positioning of the insert domain's termini is another critical factor. Proximate termini are generally advantageous, as they reduce the risk of distorting the host protein on insertion. Similarly, the selection of linkers at the receptor-effector boundaries can strongly affect the tolerance for an inserted domain as well as the allosteric coupling between receptor and effector parts. Here, we used small symmetric linkers (glycine-serine or glycine-proline-serine), as they offer a balance between facilitating receptor insertion and maintaining effective allosteric coupling, as previously observed in multiple independent contexts[6,9,36,37]. Thoroughly optimizing the linkers by systematically varying their length and composition is a promising route for further optimizing protein switches emerging from our ProDomino pipeline.

Future work could explore integrating structure-based tools such as LooDo[38] into our pipeline to further refine and optimize domain insertions predicted by ProDomino, particularly in cases where structural data are available. On top, we envision that ProDomino, in combination with recently developed methods for the de novo design of state-switching protein domains[39,40], could even support the engineering of switchable effectors based on entirely artificial and customized receptors.

Finally, as a personal note, we mention that the experimental data presented here, from the cloning of all switchable protein candidates through the entire experimental validation including the establishment of various different assays, were acquired within only ~6 months on completion of model training. This represents a major acceleration as compared to our previous protein switch engineering efforts, which often took many months to succeed even for a single, small effector protein. Taken together, we are convinced that ProDomino will enable scientists to more effectively engineer allosteric proteins in the future.

## Online content

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

# Methods

## Dataset generation

To generate a domain insertion dataset derived from natural protein sequences, the Interpro database[24,25] was downloaded and filtered for CATH domains (version from 29 June 2023). These entries were intersected with UniRef50 (ref. 26) (version from 28 June 2023), which was prefiltered for proteins <2,048 amino acids in length. The dataset was subsequently scanned for domain annotations located within another domain annotation, resulting in a total of 174,872 protein entries (intradomain insertions).

To generate datasets for model training, protein sequences were further processed by deleting the annotated insert domain sequence (parent_only). The data were then split in a 70:20:10 ratio to generate training, validation and test sets using three alternative approaches as follows. First, data were split randomly, that is at a 50% sequence identity cutoff. Second as an alternative, the dataset was split by CATH superfamily, meaning that members of a particular parent-insert domain superfamily combination only occur in the same set, that is the training, validation or test set. This ensures that the model is only validated on previously unseen protein topologies. Third, only a single example of each parent-insert domain combination was included in the data split. This strictest approach resulted in training and validation sets consisting of only 174 and 46 highly diverse protein sequences, respectively. A common intersected test set was constructed to allow side-by-side comparison of models trained using all three data splitting strategies.

## Generation of sequence embeddings

Sequence embeddings were generated by passing the amino acid sequences of all parent_only proteins in the dataset through ESM-2 (ref. 16) (https://github.com/facebookresearch/esm). We used the 3 billion-parameter version of ESM-2, which results in embeddings with a depth of 2,560 dimensions. As a baseline, we created one-hot encodings of the sequences using all canonical amino acids.

## Model architecture and training

All models were implemented in Python v.3.1 using pytorch v.2.1.0 and trained using pytorch lightning v.2.3.0. As decoder, we implemented a two-layer perceptron with rectified linear activation units. As an alternative, we also implemented BERT-style transformers with 36.7 million and 146 million parameters, respectively[41]. The Adam optimizer was used for training with default settings and a learning rate of $1 \times 10^{-5}$ (ref. 42). All models were trained on the ESM-2-derived sequence embeddings. The task was to predict, for each sequence position, whether domain insertions are tolerated or not, as evaluated by cross-entropy loss. Since domain boundaries cannot always be resolved accurately, and experimental results showed that stretches of consecutive amino acids often allow insertions, we included the two amino acids adjacent to an insertion site as additional positive labels. To account for the ambiguity of most unknown positions, we implemented sequence-specific masking. In each iteration, the known positive insertion site with its adjacent residues and one randomly selected site (negative label) were used to update the parameters in the model.

Model training and analyses was performed either on a local high-performance computer equipped with two 24-GB Nvidia RTX 4090 graphical processing units and 128 GB of RAM, or on the BwForHelix Cluster on a single A100 or A40 graphical processing unit with up to 80 GB of RAM.

## Sequence conservation, SASA and secondary structure prediction

To correlate model predictions with sequence conservation, multiple sequence alignments were computed for selected proteins using MMseq2 (https://github.com/soedinglab/MMseqs2) with parameters and databases as in colabfold[43,44]. Solvent accessible surface area (SASA) was calculated using the Biopython (version 1.81) implementation of the Shrake–Rupley algorithm with default settings[45]. Secondary structure prediction was performed using S4pred[46] (https://github.com/psipred/s4pred).

## Validation against published datasets and models

Previously published experimental domain insertion data for AraC[10], Cas9 (ref. 32) and Kir2.1 (ref. 11) were compared with the ProDomino predictions. For AraC, all positions with positive enrichment scores for at least one of the five insert domains previously screened experimentally[10] were considered insertion permissive, while all other positions were labeled as negative. For the Cas9 transposon insertion data, the same threshold was applied, but all positions for which no experimental data was available were marked as unknown. For Kir2.1, the true positive and true negative sites were identified based on Fig. 2 in ref. 11 provided they tolerated or did not tolerate at least two of the three inserts tested in the study, respectively. To benchmark the model against domain inlaid CRISPR effectors, lead insertion sites in different Cas9 orthologs that were successfully used to construct deaminase domain inlaid base editors were gathered from the literature[28–31]. The ProDomino-predicted tolerance for domain fusion at the corresponding insertion site, normalized to the mean prediction score for the respective Cas9 ortholog, was calculated.

For model benchmarking against previously reported gradient boosting classifiers[10] (https://github.com/Niopek-Lab/DI_screen), models were trained on increasing numbers of randomly selected experimental datapoints of AraC and *Spy*Cas9 and performance was tested on the remaining data. For side-by-side comparison, ProDomino was fine-tuned on the identical number of experimental data points.

## Molecular cloning

The plasmids generated and used in this study are listed in Supplementary Table 1 and annotated plasmid maps of the lead constructs are provided as GenBank files. The amino acid sequences of the relevant proteins and domains are listed in Supplementary Table 2. All constructs were generated by either standard restriction ligation cloning or Golden Gate assembly[47]. Insertion variants were generated by opening the plasmid encoding the parent protein at the insertion site by PCR, while insert domains were amplified using primers encoding the linkers as 5′ overhangs. Unless indicated otherwise in the figure legends, *As*LOV2 domain inserts were flanked by SG linkers and GR2 inserts by glycine-proline-serine linkers. Oligonucleotides were purchased from Merck and double-stranded DNA fragments were ordered from Integrated DNA Technologies. PCRs were performed using Q5 Hot Start high-fidelity DNA polymerase (New England Biolabs, cat. no. M0494L) according to the manufacturer's protocol. PCR products were analyzed by agarose gel electrophoresis followed by excision of the correct bands and DNA was purified using the QIAquick Gel Extraction Kit (Qiagen, cat. no. 28704). Restriction digests, ligations and one-pot Golden Gate reactions were performed using enzymes and buffers purchased from New England Biolabs and Thermo Fisher Scientific. Chemically competent TOP10 *E. coli* cells (Thermo Fisher Scientific, cat. no. C404003) were transformed with the assembled constructs. Finally, plasmid DNA was purified using the QIAprep Spin Miniprep or Plasmid Plus Midi kit (Qiagen, cat. nos. 27104 and 12943, respectively). All constructs were verified by Sanger sequencing using Microsynth Seqlab.

The *Renilla* luciferase expressing plasmid was obtained from Promega (pRL-TK, cat. no. E2241). pCAG-hMb3Cas12a-NLS(nucleoplasmin)-3xHA (RTW2500) was a gift from J. Bondy-Denomy, K. Joung and B. Kleinstiver (Addgene plasmid no. 115142; http://n2t.net/addgene:115142; RRID:Addgene_115142). The CAG promoter present in this construct was replaced by a cytomegalovirus promoter via Golden Gate cloning and the cytomegalovirus-driven Cas12a vector was used for all experiments in this study.

## Blue light setups

A custom-built light-emitting diode (LED) setup was used to illuminate bacterial liquid cultures. The samples were illuminated by eight high-power LEDs (type CREE XP-E D5-15, emission peak ~460 nm, emission angle ~130°, LEDTECH.DE) at a wavelength of ~460 nm. The LEDs were mounted on an aluminum plate and microtiter plates were illuminated from above at a distance of ~30 cm in a conventional shaking incubator (Minitron, Infors). The LEDs were connected to a switching mode power supply (Manson, cat. no. HCS-3102) and the current was adjusted so that samples were constantly illuminated with a light intensity of 50 µmol m$^{-2}$ s$^{-1}$.

For illumination of the agar plates, another setup consisting of an array of 96 blue LEDs (LB T64G-AACB-59-Z484-20-R33Z, Osram, emission peak 469 nm, viewing angle 30°, Mouser Electronics) mounted on a circuit board and powered by a switching mode power supply (Manson, cat. no. HCS-3102) was used. The plates were continuously illuminated at an intensity of 15 µmol m$^{-2}$ s$^{-1}$ in an incubator at 37 °C.

The setup for mammalian cell illumination was similar. Plates were placed in a humidified cell culture incubator (Eppendorf) and illuminated from below through the transparent well-plate bottom as we were working with adherent cells. A replicate control plate was incubated in the same incubator but kept in the dark. The power supply was connected to a Raspberry Pi running a custom Python script that maintained an illumination duty cycle of 5 s of light followed by 10 s darkness. Light intensity was set to 30 µmol m$^{-2}$ s$^{-1}$.

## Cell culture

HEK293T cells were cultured in 1× DMEM without phenol red (Thermo Fisher Scientific, cat. no. 31053028) supplemented with 10% (v/v) bovine fetal calf serum (Thermo Fisher Scientific, cat. no. A5670701), 2 mM L-glutamine, 100 U ml$^{-1}$ penicillin and 100 µg ml$^{-1}$ streptomycin (all Thermo Fisher Scientific, cat. nos. 25030081 and 10378016). Cells were grown at 37 °C and 5% CO$_2$ in a humidified incubator and passaged at a cell density below 80%. The cell line was authenticated before use and regularly tested for mycoplasma contamination.

## Puromycin resistance assay

HEK293T cells were seeded in 96-well plates (Starlab, cat. no. CC7682-7596) at a density of 12,500 cells per well. The next day, cells were transfected using Lipofectamine 2000 (Thermo Fisher Scientific, cat. no. 11668027) according to the manufacturer's protocol. 100 ng of each PAC fusion protein encoding construct or respective control plasmids encoding only wild-type PAC or enhanced green fluorescent protein (as negative control) were transfected per well in triplicates. Then 16 h posttransfection, the cells were challenged with 5 µg ml$^{-1}$ puromycin (Sigma-Aldrich, cat. no. P8833-10MG). Next, 72 h posttransfection, the samples were tested for cell viability using the MTT Cell Proliferation Assay Kit (Cayman Chemicals, cat. no. 10009365) according to the manufacturer's protocol.

For the microscopy assay, cells were transfected identically, but puromycin was added at a concentration of 10 µg ml$^{-1}$ 24 h posttransfection. Concurrent with the start of puromycin treatment, cells were either illuminated for 48 h or kept in the dark, followed by imaging using a Keyence BZ-9000 microscope and the Keyence BZ-II Viewer (Keyence, version 1.01) and BZ-II Analyzer software (Keyence, version 1.01).

## Chloramphenicol resistance assay

Chemically competent TOP10 E. coli were transformed with plasmids encoding a constitutively expressed CAT or insertion variants thereof and an ampicillin resistance cassette as a selection marker. Glycerol stocks were prepared from single colonies. Precultures were inoculated from the glycerol stocks and incubated overnight at 37 °C and 220 rpm in Luria-Bertani (LB) media supplemented with ampicillin at a concentration of 100 µg ml$^{-1}$. The next day, main cultures were prepared in technical triplicates in 96-well plates. 200 µl of LB media

supplemented with 100 µg ml$^{-1}$ ampicillin and the indicated concentration of chloramphenicol were inoculated with 4 µl of the precultures and incubated at 37 °C and 220 rpm. For blue light experiments, two identical culture plates were prepared, one of which was illuminated with blue light. The second plate (control) was kept in the same incubator but covered from illumination. Samples were incubated for 7 h, followed by measurement of OD600 in a Tecan infinite 200 plate reader.

## Photomask experiment

The top ten E. coli cells were cotransformed with a construct expressing the respective CAT-LOV2 fusion or an empty backbone as control together with a second kanamycin-resistant plasmid encoding an arabinose-inducible mRFP for better visualization. Cells were plated on LB agar, supplemented with ampicillin (100 µg ml$^{-1}$) and kanamycin (50 µg ml$^{-1}$). Precultures were incubated in LB media supplemented with the same antibiotics at 37 °C and 220 rpm. The illumination experiment was performed in 5 cm Petri dishes on LB agar supplemented with ampicillin (100 µg ml$^{-1}$) and kanamycin (50 µg ml$^{-1}$). A preculture was diluted 1:50 with warm top agar (0.6% LB agar) containing 16 mM arabinose and 25 µg ml$^{-1}$ chloramphenicol and poured onto the agar plate. The Petri dish was placed on an LED array and illuminated from below for 20 h while incubating at 37 °C. Images were acquired under ultraviolet (UV) light.

## Luciferase assays

HEK293T cells were seeded in 96-well plates at a density of 12,500 cells per well. To characterize dSpCas9/VPR-LOV2 hybrids, cells were transfected with a total of 100 ng of DNA using Lipofectamine 3000 (Thermo Fisher Scientific, cat. no. L3000015) according to the manufacturer's protocol 24 h after seeding. Specifically, the transfected DNA consisted of 20 ng of a firefly luciferase reporter preceded by 13 TetO sites, 20 ng of a construct encoding a Renilla luciferase and a TetO targeting sgRNA, and 40 ng of a construct encoding the respective dCas9-VPR variant. DNA was topped up to 100 ng using pBluescript as an empty stuffer plasmid. Experiments were performed in two replicates, one of which was kept in the dark, whereas the other was illuminated with blue light starting 6 h after transfection. At 48 h posttransfection, cells were lysed and firefly and Renilla luciferase activity was measured in a plate reader (Tecan Infinite 200 with i-control software (Tecan, version 2.0)) using the Dual Glo Luciferase Assay System (Promega, cat. no. E2940). Injection of each luciferase substrate was followed by a 2 s settling time and 8 s signal integration. Finally, firefly luciferase photon counts were normalized to Renilla luciferase photon counts.

To test for domain insertion tolerance in MbCas12a, cells were transfected using Lipofectamine 2000. For the PDZ domain insertion screen, a total of 200 ng of DNA per well was transfected, consisting of (1) 10 ng Renilla luciferase reporter plasmid, (2) 20 ng firefly luciferase reporter construct, (3) 85 ng sgRNA encoding construct and (4) 85 ng of a plasmid expressing the respective MbCas12a variant. For the GR2 insertion experiments, only 10 ng of the MbCas12a encoding plasmid were used, respectively, and the transfection mix was topped up with a stuffer plasmid to a total of 200 ng of DNA. Transfections were performed in triplicates. One replicate of all samples was treated with 333 µM cortisol starting 2 h posttransfection, while the control replicate was treated with dimethylsulfoxide (DMSO, solvent) at the same time point. At 48 h posttransfection, luciferase activity was measured as described above.

## Gene editing of endogenous loci

HEK293T cells were seeded in 96-well plates at a density of 12,500 cells per well. For chemical activation, cells were transfected with (1) 100 ng of a construct expressing MbCas12a and (2) 100 ng of a vector encoding the respective guide RNA (gRNA) using Lipofectamine 2000 according to the manufacturer's protocol. In case of LOV2-based experiments, a Cas12a:gRNA vector mass ratio of 3:1 was used instead. A nontargeting gRNA served as a negative control. Sequences of the gRNAs used in this study are listed in Supplementary Table 3. Blue light induction

and cortisol treatment were performed as described in the luciferase assay section above.

At 72 h posttransfection, media was aspirated and cells were lysed using DirectPCR lysis reagent (PeqLab) supplemented with 200 μg ml⁻¹ proteinase K (Sigma-Aldrich, cat. no. 03115879001). The target genomic locus was amplified with the primers listed in Supplementary Table 4 using the Q5 Hot Start High-Fidelity Polymerase (NEB). Editing frequencies were assessed by T7E1 assay or targeted amplicon sequencing. T7E1 assays were performed as previously described[37]. In brief, 5 μl of PCR amplicon was diluted in 20 μl of 1× NEB buffer 2, denatured at 95 °C for 5 min and gradually cooled to 25 °C in a GSX1 Mastercycler (Eppendorf) to form heteroduplexes. T7E1 (0.5 μl, NEB) was added, and samples were incubated at 37 °C for 15 min. Products were resolved on 2% TBE (Tris, borate and EDTA) agarose gels and imaged with a GelDoc system (Intas, 2.8 MP, 14-bit CCD). Editing efficiency was quantified using ImageJ (v.1.54k) by background subtraction and peak area analysis. Editing (%) was calculated as $100 \times (1 - (1 - \text{fraction cleaved})^{0.5})$, where fraction cleaved = cleavage products/(total DNA). For targeted amplicon sequencing, dual barcode primers were used for PCR amplification. Amplicons were resolved on a 2% 1× TBE agarose gel and bands of the expected size were excised and purified using the QIAquick Gel Extraction Kit (Qiagen). Illumina sequencing was performed using the GeneWiz amplicon sequencing service. Next-generation sequencing results were analyzed using Sabre (https://github.com/najoshi/sabre) and the CRISPresso v.2.0 suite (https://github.com/pinellolab/CRISPResso2)[48] to determine InDel frequencies at the targeted loci.

### Statistics and reproducibility

Independent experiments refer to samples cultivated, transfected or treated and handled independently and on different days. Reported fold changes (Figs. 3g and 4c,g–i) represent the ratios of mean values between the respective sample groups. For data analysis, Python v.3.1 using numpy v.1.26, Biopython v.1.81, seaborn v.0.13.1, matplotlib v.3.8.1, py3Dmol v.2.1 and notepook v.7.0.6 were applied.

### Reporting summary

Further information on research design is available in the Nature Portfolio Reporting Summary linked to this article.

## Data availability

Additional information, including relevant amino acid sequences, targeted genomic loci and PCR primers for InDel quantification are provided in the supplementary information. GenBank files of DNA constructs are provided as supplementary data file. Important constructs will be shared on Addgene (Addgene IDs 231574–31578). Raw experimental data are available on GitHub at https://github.com/Niopek-Lab/ProDomino. Protein structures for phosphoglycerate kinase (Protein Data Bank (PDB) ID 4NG4), Rvb1/Rvb2 heterohexamer (RuvB-like 1, PDB 5OAF), PAC (PDB 7K0A), CAT (PDB 1PD5), SpCas9 (PDB 4UN3) and MbCas12a (PDB 6IV6) were previously reported by others and are available on the RCSB PDB (https://www.rcsb.org/).

## Code availability

The ProDomino model is available on GitHub at https://github.com/Niopek-Lab/ProDomino.

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

## Acknowledgements

We thank all members of the Niopek laboratory and Mathony group, C. Herrmann (Heidelberg University), as well as B. E. Correia (EPFL, Lausanne) for helpful discussions. This work was supported by the European Union (ERC, DaVinci-Switches, project number 101041570). Views and opinions expressed are, however, those of the author(s) only and do not necessarily reflect those of the European Union or the European Research Council Executive Agency. Neither the European Union nor the granting authority can be held responsible for them. We also acknowledge support by the state of Baden-Württemberg through bwHPC and the German Research Foundation (DFG) through grant no. INST 35/1597-1 FUGG. We gratefully acknowledge the data storage service SDS@hd supported by the Ministry of Science, Research and the Arts Baden-Württemberg (MWK) and the German Research Foundation (DFG) through grant no. INST 35/1503-1 FUGG. D.N. is also grateful for funding by the Aventis Foundation. A.-L.v.B. is supported by the Konrad Zuse School of Excellence in Learning and Intelligent Systems (ELIZA) through the DAAD program Konrad Zuse Schools of Excellence in Artificial Intelligence, sponsored by the Federal Ministry of Education and Research.

## Author contributions

B.W., J.M. and D.N. conceived the study, designed the experiments and interpreted the data. B.W. implemented the datasets and machine learning pipeline. P.S., L.B., J.E., S.H., S.A., A.-L.v.B., N.S., A.-S.K. and J.M. designed and performed experiments. D.N. secured funding. J.M. and D.N. jointly directed the work. B.W., J.M. and D.N. jointly wrote the manuscript with the support of all authors.

## Competing interests

The authors declare no competing interests.

## Additional information

**Extended data** is available for this paper at https://doi.org/10.1038/s41592-025-02741-z.

**Correspondence and requests for materials** should be addressed to Jan Mathony or Dominik Niopek.

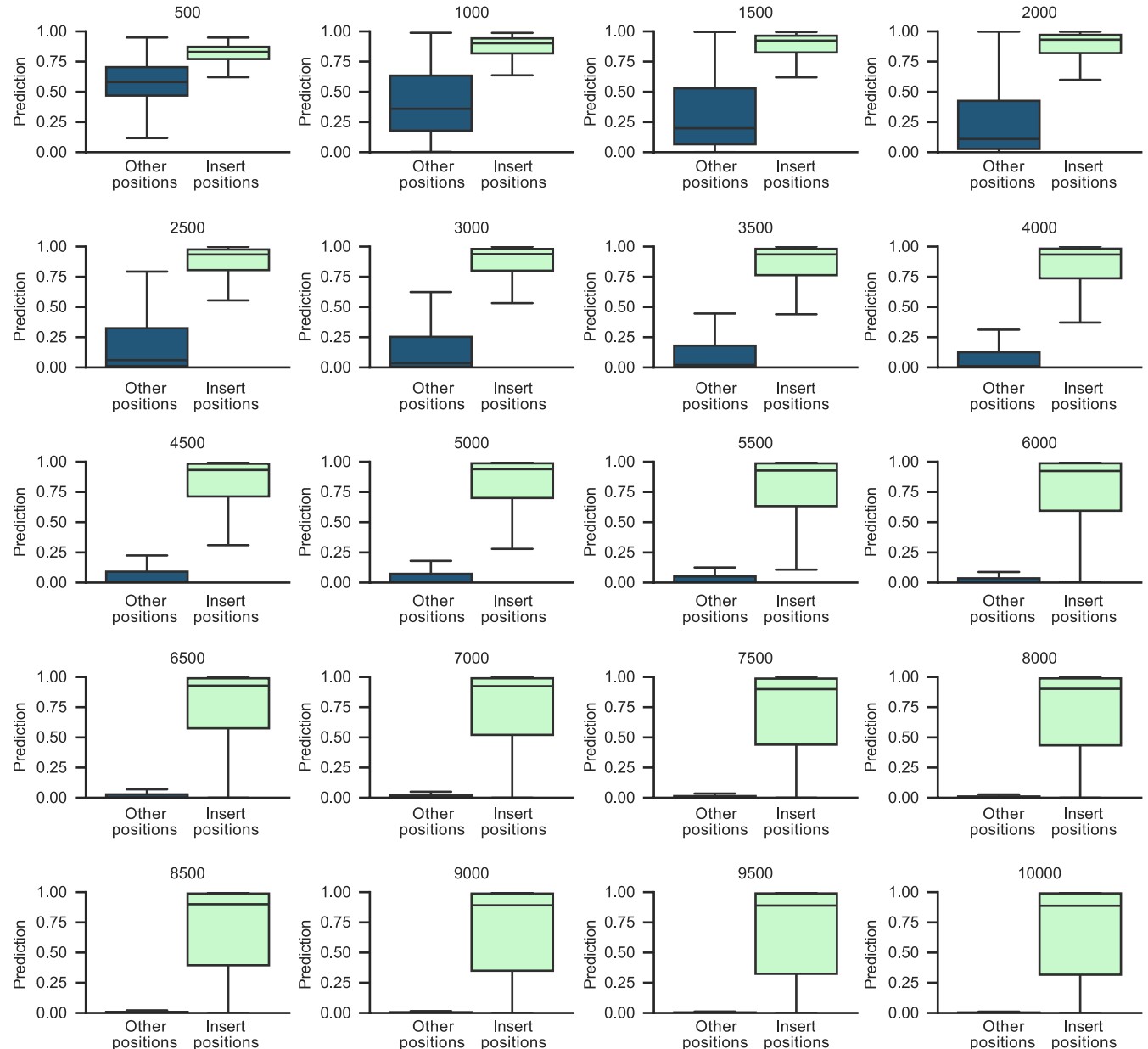

**Extended Data Fig. 1 | The number of training steps affects model sensitivity.** Model prediction scores for true insertion sites and other (unknown) positions are shown as box plots. Numbers above each plot indicate model training duration in steps. Boxes represent the interquartile range (IQR) and the median is represented by a horizontal line. Whiskers extend to the 1.5-fold IQR or to the value of the smallest or largest predicted value. n = 1,382 protein sequences with 1,382 known positive insertion sites and 325,510 unknown sites.

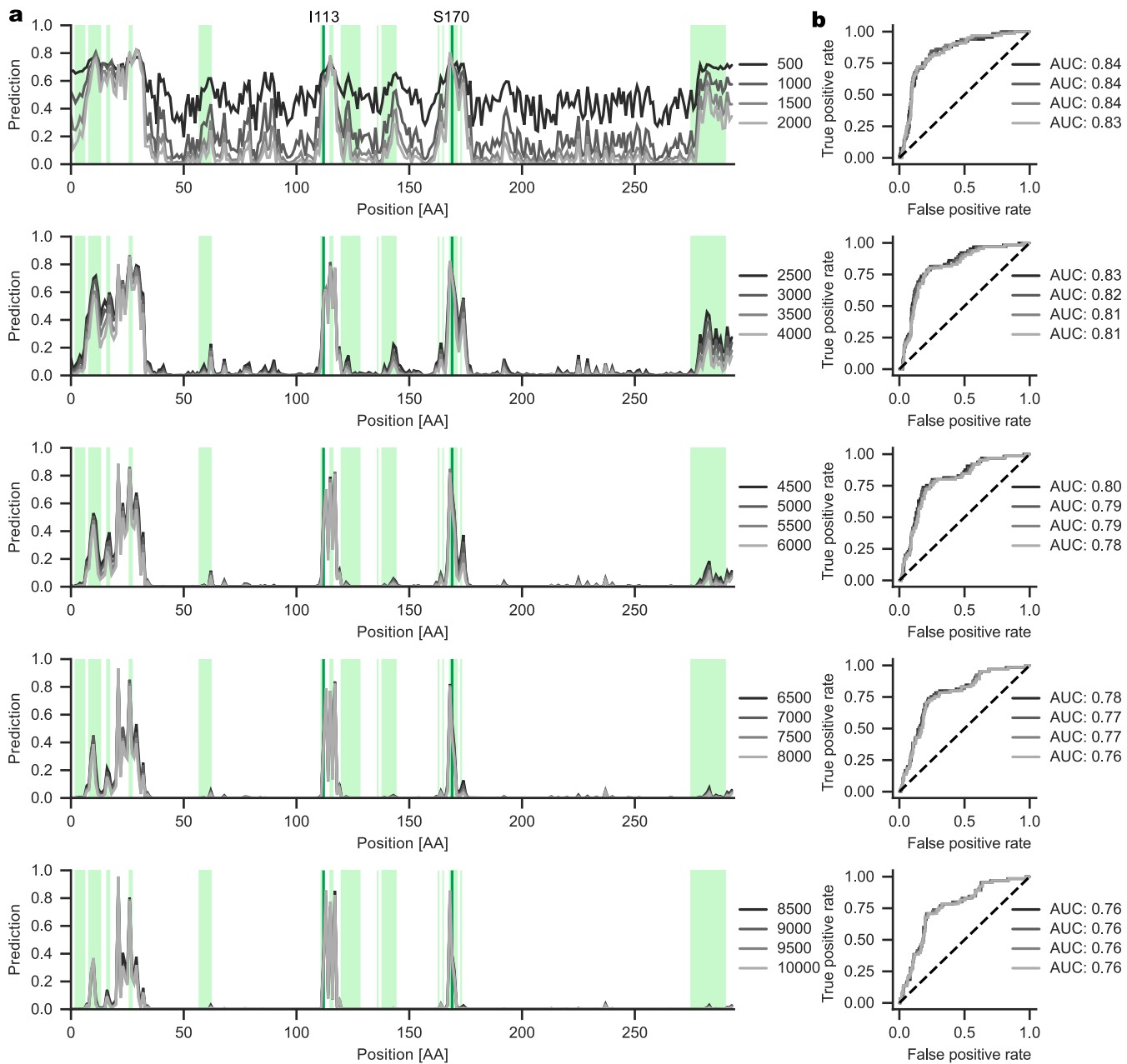

**Extended Data Fig. 2 | ProDomino correctly identifies insertion-tolerant regions in AraC. a**, The insertion score for the bacterial transcription factor AraC is shown for each amino acid position. The scores of models trained for different numbers of steps are shown in different shades of gray. Five individual subplots are presented for clarity. Green regions indicate experimentally validated insertion tolerant sites. The two sites previously used to engineer light-regulated AraC variants, I113 and S170, are indicated in dark green. **b**, ROC curves based on the predictions in **a** are shown. The area under the curve (AUC) is given for each model variant.

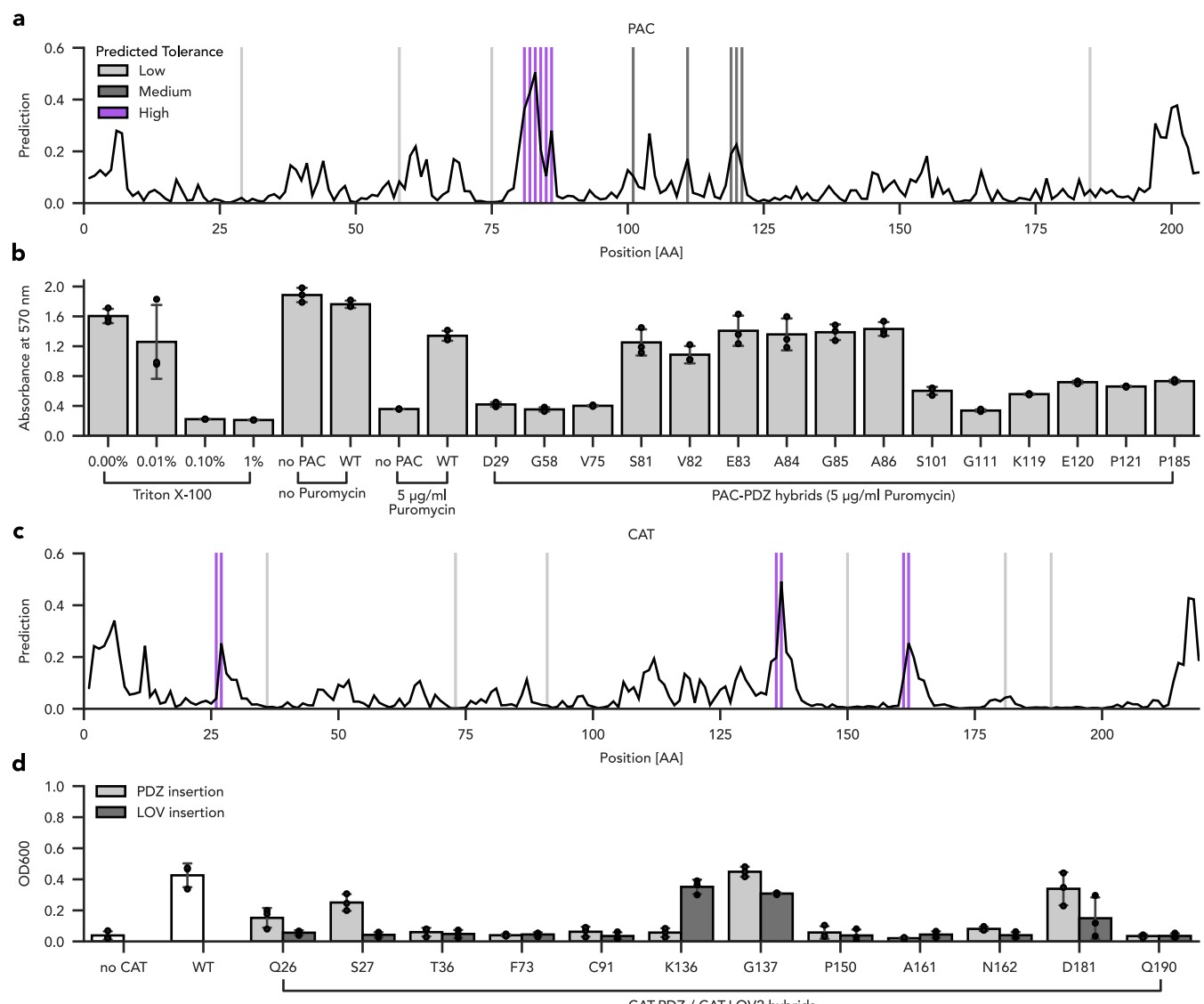

**Extended Data Fig. 3 | Domain insertion screening of PAC and CAT confirms ProDomino predictions. a, c**, ProDomino inferred insertion scores are mapped onto the primary sequence of PAC (**a**) and CAT (**c**). Insertion sites selected for experimental testing are marked by vertical lines and color coded as indicated. **b**, Assessment of insertion tolerance in PAC. HEK293T cells were transfected with vectors encoding the respective PAC variants carrying PDZ insertions after the indicated residue or a negative control expressing enhanced green fluorescent protein (eGFP). Cells were treated with 5 μg/mL puromycin and incubated for 48 hours before cell viability was assessed by MTT assay. Cells treated with different concentrations of toxic

Triton X-100 served as controls for the assay itself. Bars indicate means, error bars the standard deviation, and black dots individual data points from n = 3 independent experiments. **d**, Assessment of the CAT insertion permissibility. Bacteria were transformed with plasmids expressing the indicated CAT variant or an empty control plasmid. Liquid cultures were grown in the presence of 25 μg/mL chloramphenicol for 7 hours and cell density was assessed by measuring OD at 600 nm. Light gray bars represent PDZ insertions behind the indicated residue and dark gray bars correspond to LOV2 insertions at the same position. Bars indicate means, error bars the standard deviation, and black dots individual data points from n = 3 independent experiments.

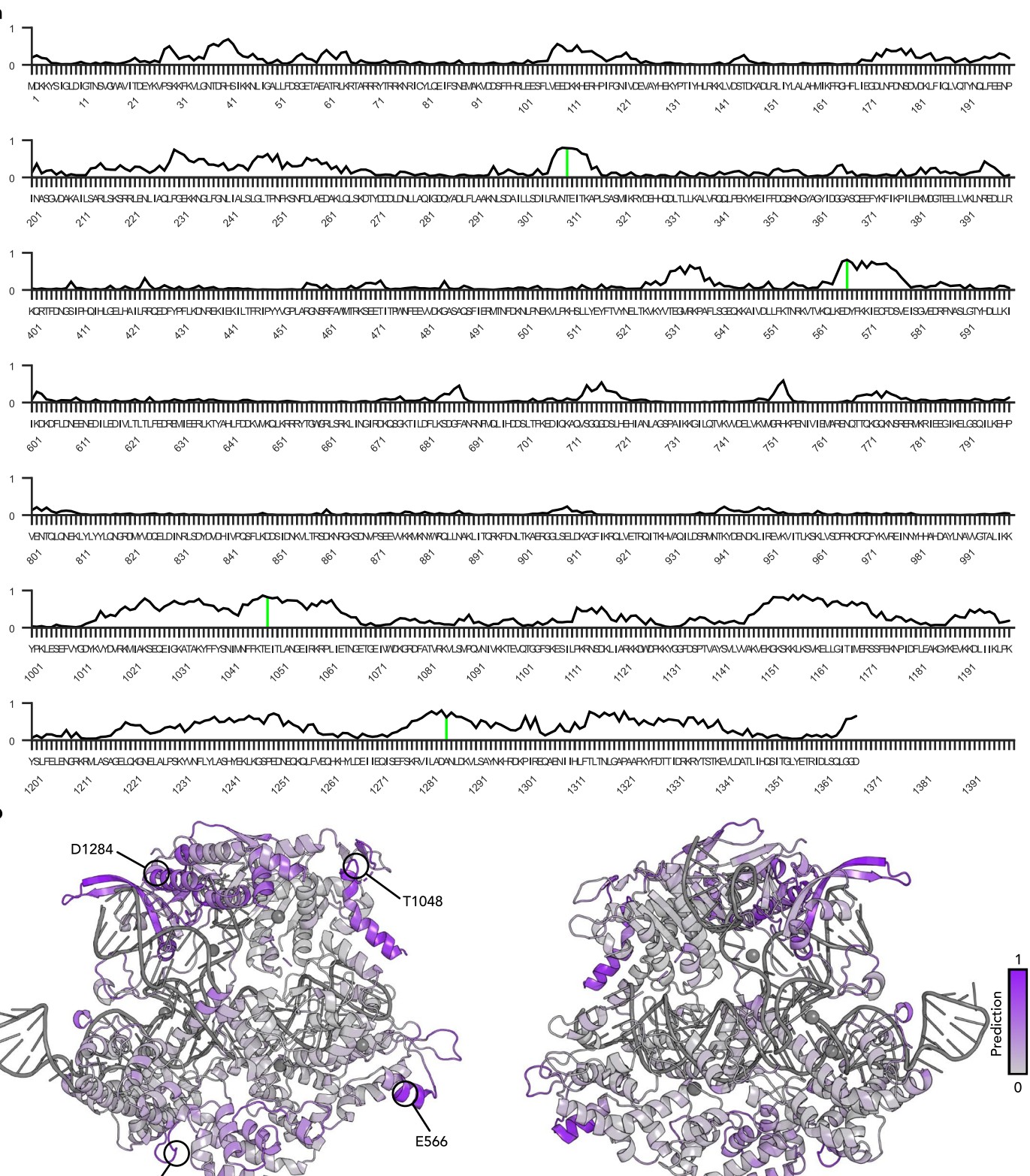

**Extended Data Fig. 4 | ProDomino prediction scores for *Spy*Cas9.** ProDomino inferred insertion scores are mapped onto the primary sequence (**a**) and the structure (**b**) of *Spy*Cas9. Insertion scores correspond to the 1,500-step model in Supplementary Fig. 13. Green indicates insertion sites selected for experimental validation in Fig. 4c. **b**, PDB ID: 4UN3.

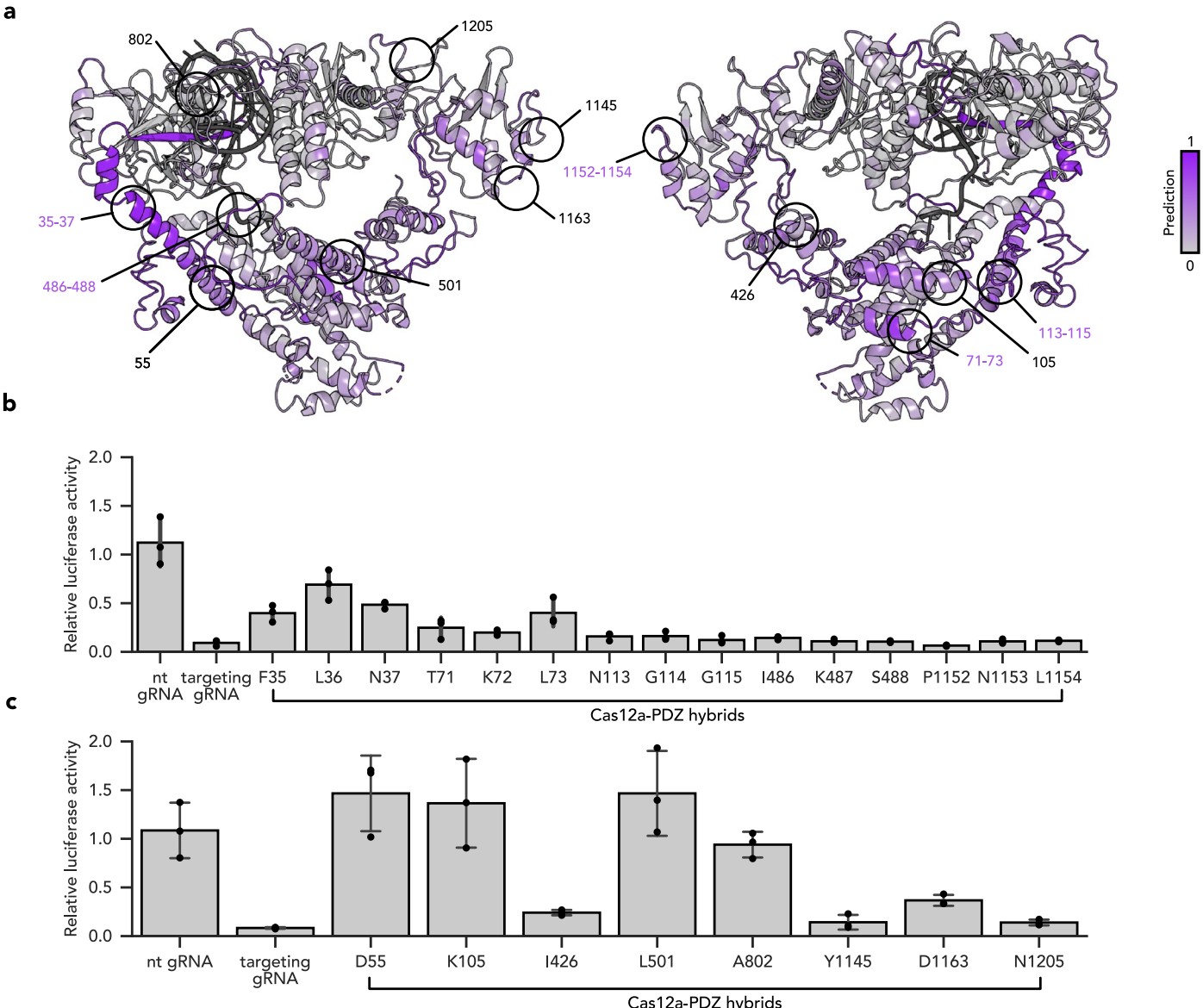

**Extended Data Fig. 5 | Experimental assessment of insertion tolerance in Cas12a. a**, Insertion scores are mapped onto a cryo-electron microscopy (cryo-EM) structure of *Mb*Cas12a. PDB ID: 6IV6. **b**, **c**, HEK293T cells were transfected with vectors encoding (i) the indicated Cas12a-PDZ insertion variant, (ii) a firefly luciferase targeting gRNA and (iii) a luciferase reporter. Samples were incubated for 48 hours, and luciferase activity was measured in a plate reader. The activity of insertion variants predicted to be active (**b**) or inactive (**c**) is shown. Bars indicate means, error bars the standard deviation, and black dots individual data points from n = 3 independent experiments. nt, non-targeting gRNA.

# Reporting Summary

## Statistics

For all statistical analyses, confirm that the following items are present in the figure legend, table legend, main text, or Methods section.

| n/a | Confirmed | |
|---|---|---|
| ☐ | ☒ | The exact sample size (*n*) for each experimental group/condition, given as a discrete number and unit of measurement |
| ☐ | ☒ | A statement on whether measurements were taken from distinct samples or whether the same sample was measured repeatedly |
| ☒ | ☐ | The statistical test(s) used AND whether they are one- or two-sided *Only common tests should be described solely by name; describe more complex techniques in the Methods section.* |
| ☒ | ☐ | A description of all covariates tested |
| ☒ | ☐ | A description of any assumptions or corrections, such as tests of normality and adjustment for multiple comparisons |
| ☐ | ☒ | A full description of the statistical parameters including central tendency (e.g. means) or other basic estimates (e.g. regression coefficient) AND variation (e.g. standard deviation) or associated estimates of uncertainty (e.g. confidence intervals) |
| ☒ | ☐ | For null hypothesis testing, the test statistic (e.g. *F*, *t*, *r*) with confidence intervals, effect sizes, degrees of freedom and *P* value noted *Give P values as exact values whenever suitable.* |
| ☒ | ☐ | For Bayesian analysis, information on the choice of priors and Markov chain Monte Carlo settings |
| ☒ | ☐ | For hierarchical and complex designs, identification of the appropriate level for tests and full reporting of outcomes |
| ☒ | ☐ | Estimates of effect sizes (e.g. Cohen's *d*, Pearson's *r*), indicating how they were calculated |

*Our web collection on statistics for biologists contains articles on many of the points above.*

## Software and code

Policy information about availability of computer code

| Data collection | Luciferase data, fluorescence, absorbance and optical density data were obtained using the i-control software (Tecan, version 2.0). Microscopy images were acquired and processed with the Keyence BZ-II Viewer (Keyence, version 1.01) and BZ-II Analyzer software (Keyence, version 1.01). Our computational pipeline is based on the Uniref50 dataset (June 28, 2023) and the CATH database (June 29, 2023) |
|---|---|
| Data analysis | Data anaylsis was performed in Python 3.1 using numpy 1.26, biopython 1.81, seaborn 0.13.1, matplotlib 3.8.1, py3Dmol 2.1 and notepook 7.0.6. Machine learning models were traned using pytorch 2.1.0, lightning 2.3.0 and previously reported gradient boosting classifiers were taken from: https://github.com/Niopek-Lab/DI_screen. Protein sequence embeddings were computed using ESM-2 (https://github.com/facebookresearch/esm). Multiple sequence alignments were computed with MMseq2 (https://github.com/soedinglab/MMseqs2). Secondary structure prediction was performed using S4pred (https://github.com/psipred/s4pred). Analysis of NGS data was performed using sabre 1.0 and CRISPResso2 2.3.1 (https://github.com/pinellolab/CRISPResso2). The ProDomino model is available on Github: https://github.com/Niopek-Lab/ProDomino. |

For manuscripts utilizing custom algorithms or software that are central to the research but not yet described in published literature, software must be made available to editors and reviewers. We strongly encourage code deposition in a community repository (e.g. GitHub). See the Nature Portfolio guidelines for submitting code & software for further information.

## Data

Policy information about availability of data

All manuscripts must include a data availability statement. This statement should provide the following information, where applicable:

- Accession codes, unique identifiers, or web links for publicly available datasets
- A description of any restrictions on data availability
- For clinical datasets or third party data, please ensure that the statement adheres to our policy

Additional information, including relevant amino acid sequences, targeted genomic loci and PCR primers for InDel quantification are provided in the supplementary information. Genbank files of DNA constructs are provided as supplementary data file. Important constructs will be shared on Addgene (Add gene ID: 231574 - 231578). Additional data will be shared upon reasonable request. Protein structures for phosphoglycerate kinase (PDB IDs: 4NG4), Rvb1/Rvb2 heterohexamer (RuvB-like 1, PDB IDs: 5OAF), PAC (PDB ID: 7K0A), CAT (PDB ID: 1PD5), SpCas9 (PDB ID: 4UN3) and MbCas12a (PDB ID: 6IV6) were previously reported by others and are available on the RCSB Protein Data Bank (https://www.rcsb.org/).

## Human research participants

Policy information about studies involving human research participants and Sex and Gender in Research.

| Reporting on sex and gender | N/A |
| --- | --- |
| Population characteristics | N/A |
| Recruitment | N/A |
| Ethics oversight | N/A |

Note that full information on the approval of the study protocol must also be provided in the manuscript.

# Field-specific reporting

Please select the one below that is the best fit for your research. If you are not sure, read the appropriate sections before making your selection.

☒ Life sciences          ☐ Behavioural & social sciences          ☐ Ecological, evolutionary & environmental sciences

For a reference copy of the document with all sections, see nature.com/documents/nr-reporting-summary-flat.pdf

# Life sciences study design

All studies must disclose on these points even when the disclosure is negative.

| Sample size | No sample size calculation was performed. The sample size was determined based on pilot experiments and aligned to conventions in the field (Benman et al., Nature Methods 22, 2025, https://doi.org/10.1038/s41592-024-02572-4; Ferreira da Silva et al., Nature Biotechnology, 2024, https://doi.org/10.1038/s41587-024-02324-x) |
| --- | --- |
| Data exclusions | No experimental data were excluded from the analysis. The protein databases were filtered during pre-processing for our machine learning pipeline, e.g. filtering for sequence identity, as described in the methods section. |
| Replication | All attempts for replication were successful. The number of replicates performed is indicated in each figure legend, where applicable. |
| Randomization | No randomization was used, as samples and controls were treated side-by-side using the identical protocols and workflows for analysis. Also, the study did not involve human or animal subjects. |
| Blinding | No blinding was used, as the majority of data was automatically collected by machines and analyzed by standard workflows. |

# Reporting for specific materials, systems and methods

We require information from authors about some types of materials, experimental systems and methods used in many studies. Here, indicate whether each material, system or method listed is relevant to your study. If you are not sure if a list item applies to your research, read the appropriate section before selecting a response.

## Materials & experimental systems

| n/a | Involved in the study |
|-----|----------------------|
| ☒ | ☐ Antibodies |
| ☐ | ☒ Eukaryotic cell lines |
| ☒ | ☐ Palaeontology and archaeology |
| ☒ | ☐ Animals and other organisms |
| ☒ | ☐ Clinical data |
| ☒ | ☐ Dual use research of concern |

## Methods

| n/a | Involved in the study |
|-----|----------------------|
| ☒ | ☐ ChIP-seq |
| ☒ | ☐ Flow cytometry |
| ☒ | ☐ MRI-based neuroimaging |

# Eukaryotic cell lines

Policy information about cell lines and Sex and Gender in Research

| | |
|---|---|
| Cell line source(s) | HEK 293T cells were obtained from ATCC (cat. no. CRL-3216) |
| Authentication | The cell line was authenticated via Single Nucleotide Polymorphism Profiling using a commercially available service (Multiplexion, Heidelberg, Germany) prior to usage. |
| Mycoplasma contamination | The cell line was tested negative for mycoplasma contamination via a commercially available service (Multiplexion, Heidelberg, Germany) prior to usage. |
| Commonly misidentified lines (See ICLAC register) | HEK (293T) is a standard cell line widely used for transient transfection and CRISPR/Cas9 experiments. |

