## [Peer Review File · Nature Methods]

Rational engineering of allosteric protein switches by in silico prediction of domain insertion sites

Corresponding Author: Professor Dominik Niopek

Version 0:

Decision Letter:

21st Feb 2025

Dear Professor Niopek,

Your Article, "Rational engineering of allosteric protein switches by in silico prediction of domain insertion sites", has now been seen by 2 reviewers. As you will see from their comments below, the reviewers find your work of considerable potential interest! They have raised a few concerns - we are interested in the possibility of publishing your paper in Nature Methods, but would like to consider your response to these concerns before we reach a final decision on publication.

We therefore invite you to revise your manuscript to address these concerns.

Link Redacted

We hope to receive your revised paper within 6 weeks. If you cannot send it within this time, please let us know. In this event, we will still be happy to reconsider your paper at a later date so long as nothing similar has been accepted for publication at Nature Methods or published elsewhere.

OPEN SCIENCE REQUIREMENTS

REPORTING SUMMARY AND EDITORIAL POLICY CHECKLISTS

IMAGE INTEGRITY

EXTENDED DATA FIGURES

DATA AVAILABILITY

All novel DNA and RNA sequencing data, protein sequences, genetic polymorphisms, linked genotype and phenotype data, gene expression data, macromolecular structures, and proteomics data must be deposited in a publicly accessible database, and accession codes and associated hyperlinks must be provided in the "Data Availability" section.

CODE AVAILABILITY

Please include a “Code Availability” subsection in the Online Methods which details how your custom code is made available. Only in rare cases (where code is not central to the main conclusions of the paper) is the statement “available upon request” allowed (and reasons should be specified).

MATERIALS AVAILABILITY

SUPPLEMENTARY PROTOCOL

To help facilitate reproducibility and uptake of your method, we ask you to prepare a step-by-step Supplementary Protocol for the method described in this paper. We [encourage authors to share their step-by-step experimental protocols](https://www.nature.com/nature-research/editorial-policies/reporting-standards#protocols) on a protocol sharing platform of their choice and report the protocol DOI in the reference list. Nature Portfolio's protocols.io is a free-to-use and open resource for protocols; protocols deposited onto protocols.io are citable and can be linked from the published article. More details can found at [protocols.io](https://www.protocols.io/help/publish-articles).

ORCID

Nature Methods is committed to improving transparency in authorship. As part of our efforts in this direction, we are now requesting that all authors identified as ‘corresponding author’ on published papers create and link their Open Researcher and Contributor Identifier (ORCID) with their account on the Manuscript Tracking System (MTS), prior to acceptance. This applies to primary research papers only. ORCID helps the scientific community achieve unambiguous attribution of all scholarly contributions. You can create and link your ORCID from the home page of the MTS by clicking on ‘Modify my Springer Nature account’. For more information please visit www.springernature.com/orcid.

Sincerely,
Arunima

Arunima Singh, Ph.D.
Senior Editor
Nature Methods

Reviewers' Comments:

Reviewer #1 (Remarks to the Author):

Summary

Wolf et al, carry out a beautiful and impressive study surveying the distributions of domain insertions across the proteome than using this database to train a machine learning model, ProDomino, to engineer switchable proteins. They use this ProDomino to enable one-shot prediction of switchable variants of several proteins. Initially, they benchmark the model on existing large-scale datasets and then harness it to engineer new variants of Cas9 and Cas12. What makes this study so impressive is that domain insertion based switch engineering is incredibly challenging. While some rationale frameworks were proposed (Dagliyan, Science, 2016) they have unfortunately not turned out to be as straightforward for distinct proteins. As a result much of the field has resolved to using brute-force library-based approaches however these are technically challenging and laborious... This machine learning-guided approach is elegant and impressive. Truly one of the most impressive uses of machine learning I have seen for protein engineering.

To me, the main question long-term is to see how well it generalizes across distinct protein classes. Honestly, I could hardly want more in a manuscript beyond more of everything... which is outside of the scope of the manuscript. For instance, I loved Figure 1 and would love a deeper survey, but I believe that to be outside of the scope of this landmark study...

Major Points

Strengths

1. The authors develop a purely sequence-based language model that shockingly can successfully predict a switchable protein's positions. Typically this style of protein engineering is so challenging and unintuitive that most in the field have switched to taking brute-force screening approaches. That they successfully predict sites that can engineer switchable proteins in an antibiotic resistance protein and Cas9/Cas12 is pretty extraordinary. Truly awesome work! Especially in the context of a publishing landscape with papers on machine learning sequence-function models that don't yield actionable predictions.

Areas of improvement:

1. The authors use a machine learning model, ProDomino, trained on natural sequences to predict domain insertion sites that will be tolerated and switchable. This same group has previously taken (Mathony, 2023, Advanced Research) a more brute force approach to generate training data for a machine learning model. I think it would be interesting to see a direct comparison for how their models perform directly on existing experimental data. I expect their purpose built models would likely perform better but I still think it would be quite interesting for the sake of comparison.

Minor Points:

1. In the last paragraph of the introduction, the authors refer to their machine learning model as deep learning. It is a language model whereas deep learning typically refers to a neural net. I recommend the authors change the language accordingly.

2. The authors compare how frequently different donor and recipient domains are associated with different domains and how similar the position of an insertion site are.

3. It's very cool to see the survey of how frequently different donor and recipient domains are associated with different domains. It would be nice to also have a comparison of what type of secondary structural elements are at this interface. I would anticipate these are mostly loops. Is there anything meaningful to glean about the properties of the insertion position?

4. The authors benchmark the model on a dataset they had previously generated on the transcription factor AraC. It would be interesting to see how it performs in other datasets that other labs have generated. For instance, they do Cas9 later but there are additional domain insertion experiments that have been done in the field. I think this would help in understanding how general and extendable the model is.

5. The authors generated single insertion variants of the lightswitchable domain Lov2 into antibiotic proteins PAC and CAT. Figure 3G covers CAT switchable experiments and has 'GS and GPG' labeled, which I am guessing are distinct linkers at the insertion site, but these are not mentioned within the figure legends. It would be good to explain these in the legends.

6. After selecting the insertion variants and testing switching of the highest positions in PAC and CAT the authors compare multiple other sites in supplemental figure 11. I think if possible it would be great to incorporate these within the main figures – this would make the results more impactful and clear. I think this rigor is a huge strength of the paper and could be highlighted more!

7. The switching from the false positive to true negative within the model is pretty awesome!

8. The authors identify sites in Cas9 that were not within a previous screen that switch Cas9 when Lov2 is inserted. How do these compare to the previously designed variants?

9. Could this model be used for engineering biosensors? It's clearly outside of the scope of the manuscript for more experiments. But were the predictions compared to the biosensors from Dave Savage's lab that were the initial use-case for transposon domain insertion (Nadler, Nature Comm, 2016)?

By Willow Coyote-Maestas

Reviewer #2 (Remarks to the Author):

SUMMARY

Wolf et al. present ProDomino, a deep learning pipeline for predicting protein domain insertion sites trained on a semi-synthetic dataset inspired by natural intradomain insertions. The model precisely identifies domain insertion sites and

effectively guides the engineering of switchable protein variants, including CRISPR-Cas systems (light- and drug-based activation), thereby accelerating protein design for research and potential clinical applications.

ASSESSMENT

This work is well-executed, and it represents an important advance in protein engineering, using machine learning techniques and a semi-synthetic dataset to develop a model that accurately predicts domain insertion sites. Its robust experimental validations across diverse proteins underscore its transformative potential, especially in creating allosteric switches. The methodological rigor shown here, as well as the broad use case of such an easy-to-use tool, make this manuscript an excellent fit for Nature Methods, as it not only advances computational methodology but also accelerates practical applications in synthetic biology.

COMMENTS

- 1) It would be great, if the authors could benchmark ProDomino vs. other existing computational tool(s) that may be currently used for domain insertion, e.g. LooDo (Rosetta), LoopGrafter or IPRO+/-.
- 2) Can the authors please elaborate on the linkers that they use for domain insertion (size, rigidity, need for individual optimization etc).
- 3) Domain insertions have been used for CRISPR base and prime editors as well, mostly to reduce gRNA-independent off-target effects or to increase modularity. Would the authors' Cas9 insertion sites also allow for the integration of a deaminase or RT domain? That would show another protein engineering use case, apart from switches.
- 4) Is there a size-restriction in terms of insertion domains? In Fig. 1e it seems as if the model was trained on short insertion domains only (given those are naturally occurring). Could that affect the efficiency of the insertion of larger domains in the engineering context?

Version 1:

Decision Letter:

Our ref: NMETH-A59100A

27th Mar 2025

Dear Dr. Niopek,

Thank you for submitting your revised manuscript "Rational engineering of allosteric protein switches by in silico prediction of domain insertion sites" (NMETH-A59100A). It has now been seen by the original referees and their comments are below. The reviewers find that the paper has improved in revision, and therefore we'll be happy in principle to publish it in Nature Methods, pending minor revisions to satisfy the referees' final requests and to comply with our editorial and formatting guidelines.

TRANSPARENT PEER REVIEW

ORCID

Sincerely,
Arunima

Arunima Singh, Ph.D.
Senior Editor
Nature Methods

Reviewer #1 (Remarks to the Author):

The authors do an excellent job improving an already stellar manuscript.

Willow Coyote-Maestas

Reviewer #2 (Remarks to the Author):

Thanks for addressing all my comments.
I congratulate the authors on this great manuscript!

Version 2:

Decision Letter:

9th May 2025

Dear Dominik,

I am pleased to inform you that your Article, "Rational engineering of allosteric protein switches by in silico prediction of domain insertion sites", has now been accepted for publication in Nature Methods. The received and accepted dates will be December 13, 2024 and May 9, 2025. This note is intended to let you know what to expect from us over the next month or so, and to let you know where to address any further questions.

Over the next few weeks, your paper will be copyedited to ensure that it conforms to Nature Methods style. Once your paper is typeset, you will receive an email with a link to choose the appropriate publishing options for your paper and our Author Services team will be in touch regarding any additional information that may be required. It is extremely important that you let us know now whether you will be difficult to contact over the next month. If this is the case, we ask that you send us the contact information (email, phone and fax) of someone who will be able to check the proofs and deal with any last-minute problems.

If you are active on Twitter/X or Bluesky, please e-mail me your and your coauthors' handles so that we may tag you when the paper is published.

Best regards,
Arunima

Arunima Singh, Ph.D.
Senior Editor
Nature Methods

** Visit the Springer Nature Editorial and Publishing website at http://editorial-jobs.springernature.com?utm_source=ejp_NMeth_email&utm_medium=ejp_NMeth_email&utm_campaign=ejp_Nmeth for more information about our career opportunities. If you have any questions please click [here](mailto:editorial.publishing.jobs@springernature.com). **

Open Access This Peer Review File is licensed under a Creative Commons Attribution 4.0 International License, which permits use, sharing, adaptation, distribution and reproduction in any medium or format, as long as you give appropriate credit to the original author(s) and the source, provide a link to the Creative Commons license, and indicate if changes were made. In cases where reviewers are anonymous, credit should be given to 'Anonymous Referee' and the source.

Reviewers' Comments:

We wish to kindly thank both reviewers for taking the time to thoroughly evaluate our work, their positive feedback and the constructive comments and suggestions, which led us to further improve our manuscript. Our answers to the reviewer's comments, detailing the changes and additions implemented during revision, are below. Of note, former Supplementary Figures 5, 6, 11, 15 and 16 are now Extended Data Figures 1-5. We look forward to your expert feedback on our revised study.

Reviewer #1 (Remarks to the Author):

Summary

Wolf et al, carry out a beautiful and impressive study surveying the distributions of domain insertions across the proteome than using this database to train a machine learning model, ProDomino, to engineer switchable proteins. They use this ProDomino to enable one-shot prediction of switchable variants of several proteins. Initially, they benchmark the model on existing large-scale datasets and then harness it to engineer new variants of Cas9 and Cas12. What makes this study so impressive is that domain insertion based switch engineering is incredibly challenging. While some rationale frameworks were proposed (Dagliyan, Science, 2016) they have unfortunately not turned out to be as straightforward for distinct proteins. As a result much of the field has resolved to using brute-force library-based approaches however these are technically challenging and laborious... This machine learning-guided approach is elegant and impressive. Truly one of the most impressive uses of machine learning I have seen for protein engineering.

We were truly delighted to receive such positive feedback on our work.

To me, the main question long-term is to see how well it generalizes across distinct protein classes. Honestly, I could hardly want more in a manuscript beyond more of everything... which is outside of the scope of the manuscript. For instance, I loved Figure 1 and would love a deeper survey, but I believe that to be outside of the scope of this landmark study...

Since we would like to put the focus really on the ML-dataset and method itself as well as the exemplary applications, we indeed would like to keep Figure 1 and the corresponding analysis largely as is. We added, in response to your comment 3 below, an analysis of the secondary structures found at the insertion sites in the protein dataset, i.e. the set that was used for model training (new Supplementary Fig. 3). Further expanding the analysis is indeed out of the scope of this work and will be followed up in subsequent studies.

Major Points

Strengths

1. The authors develop a purely sequence-based language model that shockingly can successfully predict a switchable protein's positions. Typically this style of protein engineering is so challenging and unintuitive that most in the field have switched to taking brute-force

screening approaches. That they successfully predict sites that can engineer switchable proteins in an antibiotic resistance protein and Cas9/Cas12 is pretty extraordinary. Truly awesome work! Especially in the context of a publishing landscape with papers on machine learning sequence-function models that don't yield actionable predictions.

Thank you very much for the positive comment.

Areas of improvement:

1. The authors use a machine learning model, ProDomino, trained on natural sequences to predict domain insertion sites that will be tolerated and switchable. This same group has previously taken (Mathony, 2023, Advanced Research) a more brute force approach to generate training data for a machine learning model. I think it would be interesting to see a direct comparison for how their models perform directly on existing experimental data. I expect their purpose built models would likely perform better but I still think it would be quite interesting for the sake of comparison.

We thank the reviewer for raising this interesting point. We have built ProDomino around the vision to avoid the need for experimental data to guide the identification of (allosteric) insertion sites in proteins as much as possible. In contrast, in our 2023 work by Mathony et al.¹, we have taken the opposing approach and trained gradient boosting classifiers on subsets of experimental data of insertion mutagenesis experiments to see to which extend such models (i) can extend beyond the training data and (ii) study evolutionary and biophysical constraints that may explain domain insertion tolerance. The models we had built by the time were pretty good in inferring insertion sites in a given protein only if they had seen examples of insertion variants for the same protein during training. Vice versa, the brute force model largely failed to predict domain insertion sites in proteins beyond those for which examples are revealed during model training - which arguably is the most relevant case for protein engineering. This observation motivated us to create ProDomino in the first place.

Hence, comparing ProDomino to the 2023 ML-models is somewhat like comparing apples to eggs, since one model relies on experimental data while the other does not. That said, we have now performed two types of comparative analysis, which we think are informative and underpin the particular strength of ProDomino.

First, we asked the question how many randomly selected experimental data points for AraC (our own experimental data, Mathony et al., 2023¹) or *S. pyogenes* Cas9 (Doudna/Savage transposon mutagenesis data from 2016²) we would need to reveal to the gradient boosting classifier model (Mathony et al., 2023) during training, so that it reaches the performance of ProDomino (ProDomino "baseline", i.e. not trained on any experimental data). We found that for AraC and Cas9, >20 and >300 experimental data points, respectively, are required on average, to match ProDomino's predictive power.

Secondly, we asked if the performance ProDomino would increase further, if we fine-tuned it on the identical experimental data that we employed for training the gradient boosting classifiers. We think that this comparison is, from the application perspective, more relevant since it would not make sense to ignore available experimental evidence to guide insertion site selection when using ProDomino. We observed that ProDomino outperforms the gradient

boosting classifier model for any given number of data points revealed during training/fine-tuning.

Collectively, this indicates the superior performance of ProDomino for both use cases, i.e. when no experimental data is available as well as for low-N approaches with (sparse) experimental data available for model fine-tuning.

We included these side-by-side comparisons as new Supplementary Figure 7 and describe it in new Supplementary Note 2 as follows:

“We further compared ProDomino to a machine learning approach based on gradient boosting classifier models, which we had reported in 2023 (1; we refer to it as “Mathony et al.” model). In this prior work, we had trained models on subsets of insertion mutagenesis data, i.e. these models inherently depend on experimental data.

Firstly, we asked how many randomly selected experimental data points would need to be provided to our gradient boosting classifier during training for it to reach the predictive performance of ProDomino, which operates independently of experimental data (ProDomino “baseline”). Using the available AraC¹ and *S. pyogenes* Cas9² insertion mutagenesis datasets as examples, we determined that the gradient boosting classifier models need to be trained, on average, on >20 (for AraC) and >300 (for Cas9) data points to match ProDomino’s performance (new Supplementary Fig. 7).

Next, we asked how the performance of ProDomino would improve if we revealed experimental data for a specific target protein, again using AraC and Cas9 as examples. We therefore fine-tuned ProDomino on randomly selected sets of experimental data points and evaluated model performance on the remaining data not employed for fine-tuning - akin the training of the gradient boosting classifiers. The resulting ProDomino variants consistently outperformed the Mathony et al. models, irrespective of the number of data points revealed to both models in the process (new Supplementary Fig. 7).

Supplementary Fig. 7 | Benchmarking of ProDomino against a previously published gradient boosting classifier. Gradient boosting classifiers were trained on the indicated

number of randomly selected experimental datapoints of AraC and *SpyCas9* and performance was tested on the remaining data. Datasets correspond to experimental domain insertion screen by Mathony et al.¹ (AraC) and Oakes et al.⁵ (*SpyCas9*). ProDomino baseline corresponds to the performance of standard ProDomino prior to fine tuning. Purple line indicates the performance of ProDomino when fine-tuned on the number of experimental data points identical to those used for gradient boosting classifier training. Colored shading represents the standard deviation calculated based on 50 trials of randomly selected datasets of the indicated size.

These findings underscore the robustness of ProDomino's predictive capabilities and its advantage, particularly in scenarios where experimental data is scarce or unavailable (which is the typical case). Moreover, the fact that we can boost ProDomino's performance by fine-tuning it on some experimental data points suggests that its underlying architecture enables more efficient generalization and feature learning, rendering it well-suitable for low-N engineering approaches.”

Finally, in the main text, we refer to Supplementary Note 2 as follows:

“We also compared ProDomino to a previous gradient boosting classifier model inferring domain insertion sites from experimental data and found it to be superior (Supplementary Note 2, Supplementary Fig. 7).”

Minor Points:

1. In the last paragraph of the introduction, the authors refer to their machine learning model as deep learning. It is a language model whereas deep learning typically refers to a neural net. I recommend the authors change the language accordingly.

We see the point and now consistently use the term “machine learning” instead of “deep learning” throughout the manuscript.

2. The authors compare how frequently different donor and recipient donors are associated with different domains and how similar the position of an insertion site are.

We guess this comment belongs to comment 3, right? Indeed, we address exactly this in Figure 1C.

3. It's very cool to see the survey of how frequently different donor and recipient domains are associated with different domains. It would be nice to also have a comparison of what type of secondary structural elements are at this interface. I would anticipate these are mostly loops. Is there anything meaningful to glean about the properties of the insertion position?

We thank the reviewer for this interesting question. To answer this question, we used the domain set employed for ProDomino training, for which we know the positions of the natural domain inserts. Secondary structures were predicted for the region +/- 4 residues of the domain insertion site using S4pred. We observed, as one could expect, and enrichment for

loops, which are found at ~50% of insertion sites. Interestingly, the other half are, in fact, secondary structures, about $\frac{2}{3}$ α -helices and $\frac{1}{3}$ β -sheets. Of note, we have observed in previous work that, indeed, secondary structure elements do tolerate domain fusion to a considerable extent¹, which is mirrored in this data here.

Supplementary Fig. 3 | Secondary structures around natural insertion sites. S4Pred was applied to predict secondary structure elements of the semi-synthetic proteins around the true insertion site for the “single” dataset (refer to Supplementary Note 1).

In Supplementary Note 1, we now write:

“Importantly, for all parent_only proteins, we have information about the true domain insertion site. Analysis of the secondary structure around the insertion sites revealed enrichment for coils (~50%) as compared to α -helices (~35%) and β -sheets (~15%) (Supplementary Fig. 3).”

4. The authors benchmark the model on a dataset they had previously generated on the transcription factor AraC. It would be interesting to see how it performs in other datasets that other labs have generated. For instance, they do Cas9 later but there are additional domain insertion experiments that have been done in the field. I think this would help in understanding how general and extendable the model is.

This is an excellent suggestion. Since this comment is related to comment 9, we jointly address these two comments below.

5. The authors generated single insertion variants of the lightswitchable domain Lov2 into antibiotic proteins PAC and CAT. Figure 3G covers CAT switchable experiments and has ‘GS and GPG’ labeled, which I am guessing are distinct linkers at the insertion site, but these are not mentioned within the figure legends. It would be good to explain these in the legends.

We thank the reviewer for pointing out the need to clarify the linkers used (note that reviewer 2 had a similar comment). In fact, the GS/GPG denote the linkers at the LOV2-effector boundaries. We now explicitly mention this in the Figure 3G legend as follows:

“Linkers correspond to symmetric linkers at the receptor domain boundaries. GS, Glycine-serine linker; GPG, Glycine-proline-serine linker.”

Moreover, in the Methods we now state:

“Unless indicated otherwise in the figure legends, AsLOV2 domain inserts were flanked by SG linkers and GR2 inserts by GPG linkers.”

In the discussion, we now write:

“Similarly, the selection of linkers at the receptor-effector boundaries can strongly impact the tolerance for an inserted domain as well as the allosteric coupling between receptor and effector parts. We here employed small symmetric linkers (GS or GPG), as they offer a balance between facilitating receptor insertion and maintaining effective allosteric coupling, as previously observed in multiple independent contexts^{6,9,32,33}. Thoroughly optimizing the linkers by systematically varying their length and composition is a promising route for further optimizing protein switches emerging from our ProDomino pipeline.”

6. After selecting the insertion variants and testing switching of the highest positions in PAC and CAT the authors compare multiple other sites in supplemental figure 11. I think if possible it would be great to incorporate these within the main figures – this would make the results more impactful and clear. I think this rigor is a huge strength of the paper and could be highlighted more!

Thank you for this suggestion. Since we were asked by the editor to identify relevant figures from the Supplementary Information that should be highlighted as extended data figures, we now decided to move Supplementary Figure 11 as well as Supplementary Figs. 5, 6, 15 and 16 to the extended data figures to improve their visibility.

7. The switching from the false positive to true negative within the model is pretty awesome!

Thanks!

8. The authors identify sites in Cas9 that were not within a previous screen that switch Cas9 when Lov2 is inserted. How do these compare to the previously designed variants?

The three allosteric Cas9 sites that we experimentally validated, which result in light-dependent activity upon AsLOV2 insertion (N309, E566, T1048), were not identified in previous screens. Oakes et al. reported a 4-HT-dependent Cas9 created via insertion of an estrogen receptor (ER) domain at residue 231 into Cas9. Similarly, Richter et al. reported a temperature-sensitive Cas9, which was obtained upon insertion of RsLOV domain at a similar/nearby site, i.e. behind residue N235. Interestingly, the region around residue 231-235 is also among the highly ranked positions in the ProDomino prediction, indicating that our model indeed perceives this previously validated site as candidate for insertion engineering. That said, the sites we identified (N309, E566, T1048) are distinct from that region and had even higher

ProDomino prediction scores. Moreover, we employed the AsLOV2 as insert domain, which is structurally/mechanistically very different from ER and RsLOV and, due to the proximity of its termini, particularly well-suited for domain insertion engineering. Therefore, while direct comparison is not possible due to the different receptor domains used as inserts, we can confidently say that the previously identified allosteric site (around 231-235) is in agreement with the ProDomino prediction.

9. Could this model be used for engineering biosensors? It's clearly outside of the scope of the manuscript for more experiments. But were the predictions compared to the biosensors from Dave Savage's lab that were the initial use-case for transposon domain insertion (Nadler, Nature Comm, 2016)?

We thank the reviewer for this question. We agree that biosensors are an interesting use case and it would be great to compare ProDomino predictions to the respective Nadler et al., 2016³ data set. The description (i.e. legend) of the raw data in the Supplementary Information of this paper, however, was not 100% clear regarding which position in the protein corresponds to which values (see their description of "amino_acid" in the Supplementary data 2 legend).

That said, to include additional literature data sets, we first employed the Schmidt lab data on the Kir2.1 K⁺ ion channel⁴. Well-known to the reviewer, 3 different insert peptides/domains were used to probe insertion tolerance across the entire Kir2.1 sequence in Figure 2. From this data set, we identified all sites, which well-tolerate at least two out of the three inserts (considered positive) and those that do not tolerate at least two inserts (negative). Sites with two or more neutral scores were excluded from the analysis. We observed a stark enrichment of high ProDomino scores in the set of actual positive sites according to the ground truth data.

The fact that Kir2.1 is a transmembrane protein, unlike the other soluble proteins in our study, provides independent evidence of ProDomino's general applicability in predicting domain fusion sites across diverse proteins (new Supplementary Fig. 8a).

Supplementary Fig. 8 | Cross-validation of ProDomino performance on previously published Datasets. **a**, Boxplot of ProDomino predictions for insertion sites in Kir2.1 previously published by Coyote-Maestas et al ⁴. The true positive and true negative sites were identified based on Fig. 2 in reference 2 provided they tolerated or did not tolerate at least two of the three inserts tested in the study, respectively. **b**, Insertion sites in different Cas9

orthologues that were successfully employed to construct domain inlaid CRISPR base editors were gathered from the literature⁵⁻⁸. The ProDomino-predicted tolerance for domain fusion at the corresponding insertion site, normalized to the mean prediction score for the respective Cas9 ortholog, is shown (i.e. random choice would result in a value of 1).

Secondly, we performed an analysis on the basis on domain interlaid CRISPR base editors in response to comment 3 by reviewer 2, which provides additional, independent validation for ProDomino (see below, new Supplementary fig. 8b).

In the main text, we now write:

“Finally, as additional independent benchmarking, we employed previously reported domain insertion datasets for the Kir2.1 K⁺ ion channel⁴ and engineered interlaid deaminase domain CRISPR base editors⁵⁻⁸. We observed that the ProDomino predictions matched the experimental data well overall (Supplementary Fig. 8).”

By Willow Coyote-Maestas

Reviewer #2 (Remarks to the Author):

SUMMARY

Wolf et al. present ProDomino, a deep learning pipeline for predicting protein domain insertion sites trained on a semi-synthetic dataset inspired by natural intradomain insertions. The model precisely identifies domain insertion sites and effectively guides the engineering of switchable protein variants, including CRISPR-Cas systems (light- and drug-based activation), thereby accelerating protein design for research and potential clinical applications.

ASSESSMENT

This work is well-executed, and it represents an important advance in protein engineering, using machine learning techniques and a semi-synthetic dataset to develop a model that accurately predicts domain insertion sites. Its robust experimental validations across diverse proteins underscore its transformative potential, especially in creating allosteric switches. The methodological rigor shown here, as well as the broad use case of such an easy-to-use tool, make this manuscript an excellent fit for Nature Methods, as it not only advances computational methodology but also accelerates practical applications in synthetic biology.

We kindly thank the reviewer for this very positive feedback.

COMMENTS

1) It would be great, if the authors could benchmark ProDomino vs. other existing computational tool(s) that may be currently used for domain insertion, e.g. LooDo (Rosetta), LoopGrafter or IPRO+/-.

We appreciate the reviewer's suggestion to benchmark ProDomino against existing computational tools such as LooDo, LoopGrafter, and IPRO+/- . However, we believe that a direct comparison/benchmarking is actually not possible, as these tools are designed for distinct tasks and operate under fundamentally different assumptions than ProDomino.

ProDomino is uniquely specialized for predicting viable domain insertion sites in host protein sequences, enabling the rational design of functional multi-domain proteins, particularly allosteric switches. In contrast, LooDo, LoopGrafter, and IPRO+/- serve different purposes:

LooDo (Rosetta) is a powerful tool for structural refinement and loop-based domain insertion modeling. Very importantly, however, it does not inherently predict where a domain should be inserted. Instead, it requires the user to predefine insertion sites, and it then designs and optimizes the structural integration of the inserted domain. Additionally, LooDo relies on high-resolution structural input data, whereas ProDomino is designed to work solely from sequence, which is crucial given that experimental structures are unavailable for many proteins of interest. That said, while LooDo serves a different purpose, we think that it could be interesting to use it later in the design process, to optimize hybrid proteins based on ProDomino-inferred domain insertions sites.

LoopGrafter is specifically designed for loop transplantation between homologous proteins, enabling fine-tuned modifications of structurally related enzymes or scaffolds. It is not meant for inserting large, functionally unrelated domains into new protein contexts. Furthermore, it requires structural alignments between the scaffold and donor loops, making it unsuitable for de novo domain insertions where structural information is lacking.

IPro+/- is focused on local sequence and structure modifications, particularly optimizing active sites in enzymes, ligand binding pockets, or protein interfaces through small indels. While it allows for sequence modifications beyond point mutations, it is not designed for inserting entire domains. Like the other tools, it also requires structural data, making it fundamentally different from ProDomino, which can guide domain insertion decisions purely from sequence-based predictions.

Given these distinctions, none of these tools can replace ProDomino's role in guiding domain insertion engineering, as they do not provide the key functionality of scoring potential insertion sites across a protein sequence. Rather than benchmarking ProDomino against tools designed for different use-cases, we argue that these approaches are complementary: ProDomino identifies promising insertion sites, while structure-based tools such as LooDo could be used subsequently to model and refine the inserted domain's structural integration if structural data is available. We now state in the discussion:

"Future work could explore integrating structure-based tools such as LooDo³⁴ into our pipeline to further refine and optimize domain insertions predicted by ProDomino, particularly in cases where structural data is available."

Finally, in his first comment, reviewer 1 raised a related question and asked us to compare ProDomino to our previously reported machine learning models inferring domain insertion sites on the basis of available experimental data (Mathony et al., 2023). While also this comparison is arguably rather arbitrary, since ProDomino's main strength is its independence of experimental ground truth, this analysis indicates the superior performance of ProDomino also for cases in which some experimental data is, indeed, available to guide feature learning.

2) Can the authors please elaborate on the linkers that they use for domain insertion (size, rigidity, need for individual optimization etc).

We thank the reviewer for the opportunity to elaborate on this important aspect. Indeed, the selection of linkers at the receptor-effector boundaries is well known to significantly affect both, the tolerance for an insert domain as well as allosteric coupling between receptor and effector parts of protein switches. That said, in our case, we did not perform extensive linker optimization, but simply selected small (2-3 residue) symmetric linkers, either flexible (GS) or semi-rigid (GPG). This choice was made, since we and others previously found such linkers to represent a good compromise between facilitating receptor insertion while still allowing for effective allosteric coupling in various independent contexts⁹⁻¹².

That said, the linkers could certainly be further optimized, and it is rather likely that some of the protein switches presented in our manuscript could thereby be further improved.

In the discussion, we now added a brief note on linker selection and optimization as follows:

“Similarly, the selection of linkers at the receptor-effector boundaries can strongly impact the tolerance for an inserted domain as well as the allosteric coupling between receptor and effector parts. We here employed small symmetric linkers (GS or GPG), as they offer a balance between facilitating receptor insertion and maintaining effective allosteric coupling, as previously observed in multiple independent contexts⁹⁻¹². Thoroughly optimizing the linkers by systematically varying their length and composition is a promising route for further optimizing protein switches emerging from our ProDomino pipeline.”

3) Domain insertions have been used for CRISPR base and prime editors as well, mostly to reduce gRNA-independent off-target effects or to increase modularity. Would the authors' Cas9 insertion sites also allow for the integration of a deaminase or RT domain? That would show another protein engineering use case, apart from switches.

The reviewer raises an interesting point regarding additional use cases of Prodomino, e.g. for “domain inlaid” CRISPR base or prime editor design. First of all, the reason why we trained ProDomino on a dataset derived from intradomain insertion events specifically and not just events of simple domain fusion was to bias the prediction towards sites that (i) tolerate insertions but also (ii) enable a tight functional entanglement (conformational coupling) between the insert domain (receptor) and the host protein (effector). While being an essential prerequisite for allosteric protein switches to work, we are not sure if tight conformational coupling is required (or actually even desired) for use cases such as base or prime editors with integrated effector domains. In these cases, we would think that other parameters, such as orientation of the deaminase relative to the non-target DNA strand, would be important criteria - and ProDomino is unaware of these.

That being said, we would expect that ProDomino should at least be capable of serving as a rough filter to identify sets of potential candidate fusion sites, e.g. for deaminase domains in CRISPR-Cas9, which would then be further selected for additional criteria. To investigate this hypothesis, we curated the lead domain insertion sites underlying previously reported domain inlaid base editors based on *SpyCas9*, *NmeCas9* and *SauCas9*⁵⁻⁸. We then computed their ProDomino-predicted tolerance for domain fusion relative to the mean ProDomino prediction score for the respective Cas9 ortholog (i.e. random choice would result in a value of 1). Indeed, we observed that the majority of sites successfully used previously to construct domain inlaid CRISPR-Cas9 base editors correspond to regions that ProDomino also predicted as being generally insertion-permissive (3-fold higher values than random choice, on average) (new Supplementary Fig. 8b; note that Supplementary Fig. 8a corresponds to a request from reviewer 1). This indicates that ProDomino could, in fact, serve as a first, rough filter to exclude sites that are not insertion permissive, e.g. to inform the engineering process of domain inlaid Cas9 effectors.

Supplementary Fig. 8 | Cross-validation of ProDomino performance on previously published Datasets. **a**, Boxplot of ProDomino predictions for insertion sites in Kir2.1 previously published by Coyote-Maestas et al.⁴. The true positive and true negative sites were identified based on Fig. 2 in reference 2 provided they tolerated or did not tolerate at least two of the three inserts tested in the study, respectively. **b**, Insertion sites in different Cas9 orthologues that were successfully employed to construct domain inlaid CRISPR base editors were gathered from the literature^{5–8}. The ProDomino-predicted tolerance for domain fusion at the corresponding insertion site, normalized to the mean prediction score for the respective Cas9 ortholog, is shown (i.e. random choice would result in a value of 1).

In the main text, we now write:

“Finally, as additional independent benchmarking, we employed previously reported domain insertion datasets for the Kir2.1 K⁺ ion channel⁴ and engineered interlaid deaminase domain CRISPR base editors^{5–8}. We observed that ProDomino prediction matched the experimental data well overall (Supplementary Fig. 8).”

4) Is there a size-restriction in terms of insertion domains? In Fig. 1e it seems as if the model was trained on short insertion domains only (given those are naturally occurring). Could that affect the efficiency of the insertion of larger domains in the engineering context?

It is true that the data set contains proteins with insert domains with a median size around 100 amino acids. However, the data set still contains insert domains that are larger as well (up to about 400 residues). This is well within the range of receptor domains typically used for protein switch engineering in synthetic biology.

That said, it is very true that larger domain inserts are, on average, less well tolerated than smaller ones, as we previously observed in an unbiased insertion tolerance screen using different insert domains¹. Beyond size, close proximity of N- and C-Termini of the insert domain is desired to avoid distortion of the host protein structure.

We note that our study covers three different insert domains (PDZ, LOV2 and a circularly permuted GR2), which all are well tolerated, albeit spanning a size range from 6 kDa to 30 kDa. Importantly, for all three domains, the termini are in close proximity; for GR2, we had actually circularly permuted the receptor in prior work¹⁰ to position the termini accordingly.

To make readers aware of the potential impact of insert size and termini positioning on engineering success, we now added the following section to the discussion:

“While ProDomino focuses on predicting domain insertion sites in effector proteins, the choice of the insert domain (receptor) is equally crucial for engineering success. Our previous observations suggest that employing larger insert domains tends to negatively impact insertion tolerance¹. Beyond size, the positioning of the insert domain’s termini is another critical factor. Proximate termini are generally advantageous, as they reduce the risk of distorting the host protein upon insertion.”

References:

1. Mathony, J., Aschenbrenner, S., Becker, P. & Niopek, D. Dissecting the Determinants of Domain Insertion Tolerance and Allostery in Proteins. *Adv. Sci.* **10**, 2303496 (2023).
2. Oakes, B. L. *et al.* Profiling of engineering hotspots identifies an allosteric CRISPR-Cas9 switch. *Nat. Biotechnol.* **34**, 646–651 (2016).
3. Nadler, D. C., Morgan, S. A., Flamholz, A., Kortright, K. E. & Savage, D. F. Rapid construction of metabolite biosensors using domain-insertion profiling. *Nat. Commun.* **7**, 1–11 (2016).
4. Coyote-Maestas, W., He, Y., Myers, C. L. & Schmidt, D. Domain insertion permissibility-guided engineering of allostery in ion channels. *Nat. Commun.* **10**, 1–14 (2019).
5. Bamidele, N. *et al.* Domain-inlaid Nme2Cas9 adenine base editors with improved activity and targeting scope. *Nat. Commun.* **15**, 1458 (2024).
6. Nguyen Tran, M. T. *et al.* Engineering domain-inlaid SaCas9 adenine base editors with reduced RNA off-targets and increased on-target DNA editing. *Nat. Commun.* **11**, 4871 (2020).
7. Li, S. *et al.* Docking sites inside Cas9 for adenine base editing diversification and RNA off-target elimination. *Nat. Commun.* **11**, 5827 (2020).
8. Chu, S. H. *et al.* Rationally Designed Base Editors for Precise Editing of the Sickle Cell Disease Mutation. *CRISPR J.* **4**, 169–177.
9. Dagliyan, O. *et al.* Engineering extrinsic disorder to control protein activity in living cells. *Science* **354**, 1441–1444 (2016).
10. Brenker, L. *et al.* A Versatile Anti-CRISPR Platform for Opto- and Chemogenetic Control of CRISPR-Cas9 and Cas12 across a Wide Range of Orthologs. 2024.11.25.625186 Preprint at <https://doi.org/10.1101/2024.11.25.625186> (2024).
11. Hoffmann, M. D. *et al.* Optogenetic control of *Neisseria meningitidis* Cas9 genome editing using an engineered, light-switchable anti-CRISPR protein. *Nucleic Acids Res.* **49**, 1–11 (2021).
12. Bubeck, F. *et al.* Engineered anti-CRISPR proteins for optogenetic control of CRISPR–Cas9. *Nat. Methods* **15**, 924–927 (2018).